# Thresholds of lake and reservoir connectivity in river networks control nitrogen removal

Noah M. Schmadel[1], Judson W. Harvey[1], Richard B. Alexander[1], Gregory E. Schwarz[1], Richard B. Moore[2], Ken Eng[1], Jesus D. Gomez-Velez[3], Elizabeth W. Boyer[4] & Durelle Scott[5]

Lakes, reservoirs, and other ponded waters are ubiquitous features of the aquatic landscape, yet their cumulative role in nitrogen removal in large river basins is often unclear. Here we use predictive modeling, together with comprehensive river water quality, land use, and hydrography datasets, to examine and explain the influences of more than 18,000 ponded waters on nitrogen removal through river networks of the Northeastern United States. Thresholds in pond density where ponded waters become important features to regional nitrogen removal are identified and shown to vary according to a ponded waters' relative size, network position, and degree of connectivity to the river network, which suggests worldwide importance of these new metrics. Consideration of the interacting physical and biological factors, along with thresholds in connectivity, reveal where, why, and how much ponded waters function differently than streams in removing nitrogen, what regional water quality outcomes may result, and in what capacity management strategies could most effectively achieve desired nitrogen loading reduction.

[1] U.S. Geological Survey, Reston 20192 VA, USA. [2] U.S. Geological Survey, Pembroke 03275 NH, USA. [3] Department of Civil and Environmental Engineering, Vanderbilt University, Nashville 37212 TN, USA. [4] Department of Ecosystem Science and Management, Pennsylvania State University, State College 16802 PA, USA. [5] Department of Biological Systems Engineering, Virginia Polytechnic Institute and State University, Blacksburg 24061 VA, USA. Correspondence and requests for materials should be addressed to N.M.S. (email: nschmadel@usgs.gov)

River corridors are comprised of a mosaic of lotic and lentic water body types—which store, convey, and transform mass and energy through accumulated biogeochemical processing of transported materials—from headwaters to oceans[1–3]. The lotic versus lentic composition of a river network may strongly influence biogeochemical processing, depending on specific hydrologic and biological conditions[4,5]. For this study, we refer to the myriad of lentic waters, including lakes, reservoirs, small impoundments, and wetlands, as ponded waters, and we focus specifically on ponded waters with a direct connection to the river network.

Many ponded waters serve important human needs. For example, constructed reservoirs often serve dual needs for water resources and flood control. Along with positive functions, dams may also have long-term negative impacts by trapping and accumulating nutrients and fine sediment[6–8], potentially concentrating nutrients that result in cyanobacterial blooms and the release of toxic compounds[9]. Furthermore, dams cause flow fragmentation, regulating downstream river flow[10] and affecting organism dispersal and biodiversity[11]. Conversely, other types of ponded waters, such as constructed ponds for water treatment or naturally occurring ponded waters (e.g., wetlands and beaver ponds), may have a more favorable balance of hydrologic and biological functions that benefit water quality[12–17]. Here, we focus on using widely available data and modeling to improve the physical basis for water quality models to explain where, why, and how much nitrogen is removed by ponded waters relative to the streams they replace in river networks.

Although the relative roles of streams and ponded waters may vary throughout river networks and across regions, their individual rates of nitrogen removal often vary consistently[1,13,18]. The removal of nitrogen is related to the time water spends in contact with benthic surfaces, with the proportion removed generally scaling with the water residence time and depth (the water volume to benthic surface area ratio)[19,20]. Ponded water size, which is correlated with water residence time, therefore, has frequently been identified as an important determinant of nitrogen removal[21,22]. Cumulative effects of ponded waters have been more challenging to explain. For example, ponded waters can be responsible for a majority of regional nitrogen removal[23], whereas in other basins, ponded waters may have overwhelming local effects while little overall regional effect[1]. The downstream accumulation and blending of lotic and lentic processes in river networks may obscure causes, with different explanatory metrics or sparse data on low-resolution networks sometimes hindering comparisons. A more thorough understanding of cumulative ponded water influences on nitrogen loading is needed, utilizing rich datasets from real networks to support an analysis of where and why ponded waters are more effective than streams at removing nitrogen.

Here, we identify specific relationships between hydrologic and biological factors that control local and cumulative nitrogen removal in the Northeastern United States and investigate the role of ponded waters that vary systematically in shape, size, location, and number across the region. We use predictive modeling to specify where in the network the annual removal of total nitrogen is expected to be dominated by streams or ponded waters and explain the cumulative effects on downstream waters. We use a spatially referenced water quality model that was calibrated for current conditions with comprehensive river water quality, land use, and river network hydrography datasets[24] (see Methods). The effects of pond density, size, shape, connectivity to the river network, and position in the network are evaluated across two sub-regions of the Northeastern United States (Chesapeake Bay watershed (CB) and New England (NE)).

To directly compare the role of ponded waters versus streams in nitrogen removal, we replaced all of the ponded waters (18,180 unique water bodies) in the calibrated model with appropriately sized streams using consistent estimation procedures for stream hydraulic geometry and nitrogen removal (see Methods). This alternative model without ponded waters was run in simulation mode and compared to the calibrated model with ponded waters to clarify the role of ponded waters in regional nitrogen removal, identify the physical and biological factors that best explain where, why, and how much nitrogen is removed by ponded waters relative to streams, and provide guidance that informs management priorities about where to remove (or create) dams, or to restore streams, to best manage riverine nitrogen loads.

Tradeoffs between the hydrologic factors affecting residence time and the biological activity in ponded waters determine their effectiveness relative to streams in removing nitrogen, leading to identification of thresholds in densities, and the physical controls of relative size, shape, and connectivity to the river network. We find that interactions between the physical characteristics of ponded waters and the river network determine the local and cumulative effects of ponded waters on regional loading of nitrogen to coastal areas.

## Results

**Regional patterns.** Our results showed that the effect of ponded waters on cumulative nitrogen removal—estimated from headwaters to coasts as the total amount removed upstream of each location—varies throughout the river network (Fig. 1a; see Methods). The river network with ponded waters typically removed a greater cumulative proportion of nitrogen (i.e., dominant) relative to a river network with stream replacements across both CB and NE, with some notable exceptions where the stream replacements dominated nitrogen removal (e.g., New Jersey coastal plain; Fig. 1a). Regionally, a greater fraction of total nitrogen was removed by ponded waters in NE (22.8% of total nitrogen removal occurred in ponded waters in NE versus only 5.1% in CB). Correspondingly, the pond density (defined as cumulative upstream pond surface area divided by cumulative upstream drainage area) is greater in NE compared to CB (2.9% pond density in NE versus 0.4% in CB; Supplementary Table 1). Striking regional differences confirms a need to determine how and where the management of ponded waters can reduce downstream nitrogen loading.

We quantified clear thresholds in pond density above which ponded waters start to dominate over streams in cumulative nitrogen removal (Figs. 1b, 2; see Methods). A breakdown by stream order identifies position in the network as important in explaining nitrogen removal. Our results confirm that while a ponded waters' contribution to nitrogen removal is greatest in headwaters, the blending or downstream accumulation of lotic and lentic processes translates to smaller cumulative effects of ponded waters in progressively higher-order rivers (Fig. 2). Therefore, pond density together with position in the network are good basic indicators of the cumulative effects in increasingly larger river basins. Although pond density is consistently greater in NE than in CB at each stream order (Supplementary Fig. 1), the pond density threshold is generally lower in CB (0.8% mean density threshold; Fig. 2a, c) compared with NE (2.5% mean density threshold; Fig. 2b, d and Supplementary Table 2). A lower pond density threshold suggests that it takes fewer ponded waters to cause a cumulative effect in CB. Yet the role of ponded waters in nitrogen removal decreases with increasing stream order more drastically in CB than in NE. Blending with lotic processes is more extreme in CB than in NE, obscuring the cumulative effect of ponded waters (Supplementary Fig. 2), as indicated by the

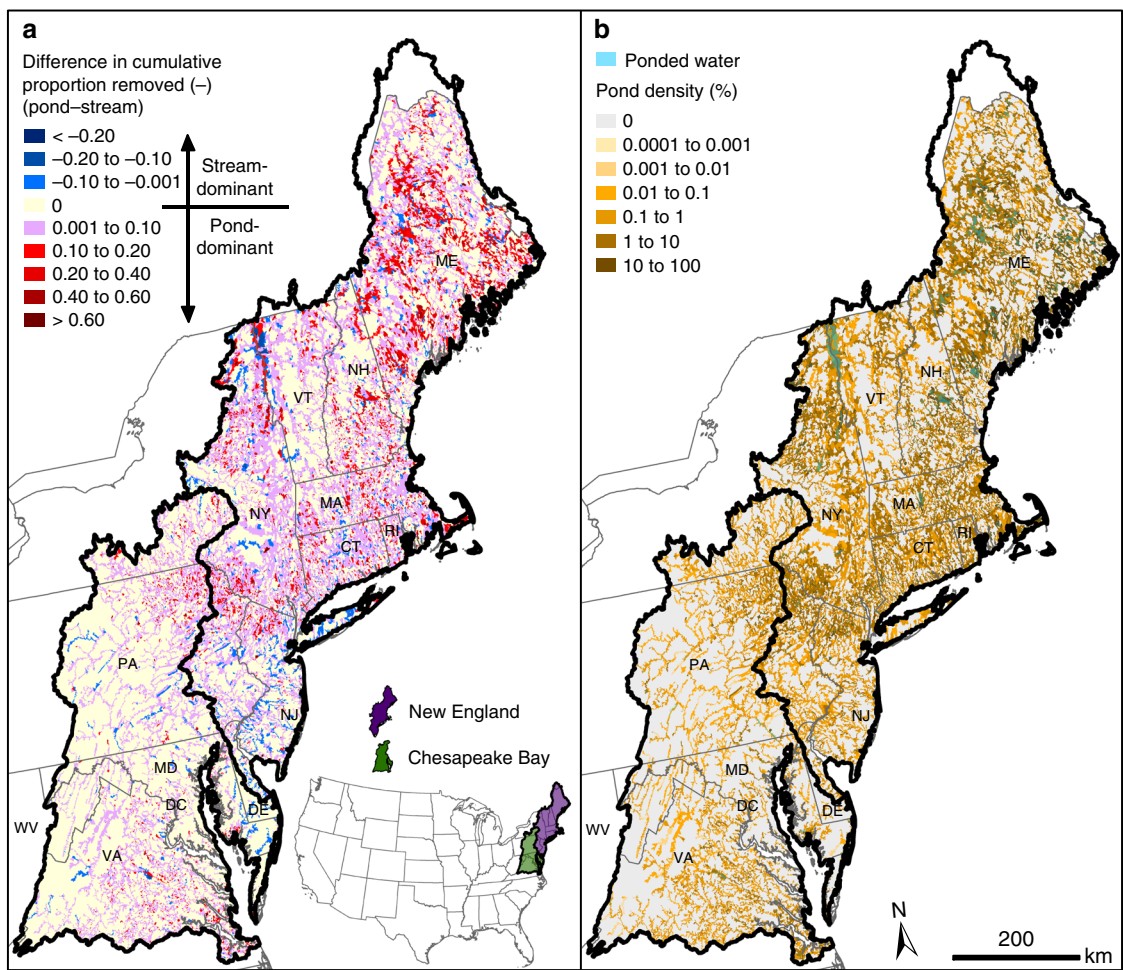

**Fig. 1** The relative effect of ponded waters versus streams on nitrogen removal in the Northeastern United States. **a** Cumulative proportion of total nitrogen removed annually by ponded waters relative to streams (expressed as a difference) throughout the river network, where each stream reach and ponded water catchment is colored. A positive (red) value indicates that nitrogen removal is dominated by ponded waters and a negative value (blue) indicates that nitrogen removal is dominated by streams (see Methods). **b** Pond density (cumulative upstream pond surface area to cumulative upstream drainage area) throughout the river network. Maps created using model results, data from NHD[35], and expressions in the Methods

cumulative stream surface area becoming greater than that of ponded waters lower in the network in CB but not in NE (Supplementary Fig. 1).

**Local factors explaining regional patterns**. We focused on the individual (local) balance of hydrologic and biological factors to better explain why headwater ponded waters have the largest influence, and why there are differences in pond density thresholds across stream order and between the two sub-regions. Whether an individual ponded water or a stream dominates nitrogen removal (i.e., removes a greater proportion) depends on the tradeoff between water residence time and biological reaction rate. The Damköhler number (Da) is the ratio of physical retention timescale to biological reaction timescale[25], which we computed using the standard hydraulic metric of time required to displace a unit volume of water (referred to as the reciprocal hydraulic load[26]). The Damköhler numbers for a ponded water and its stream replacement are:

$$Da_p = \frac{A_p}{Q_p} \cdot \nu_p \qquad (1)$$

$$Da_s = \frac{L \cdot b}{Q_s} \cdot \nu_s \qquad (2)$$

where $A_p$ is the pond surface area (m$^2$); $L$ is the stream reach length intersecting the ponded water (m); $b$ is the mean stream width (m); $Q_p$ and $Q_s$ (m$^3$ d$^{-1}$) are the mean annual discharges (assumed equal over the annual time period, so that $Q_p = Q_s$); $\frac{A_p}{Q_p}$ and $\frac{L \cdot b}{Q_s}$ are the reciprocal hydraulic loads (d m$^{-1}$); $\nu_p$ and $\nu_s$ are the nitrogen removal uptake velocities (m d$^{-1}$) and measures of biological activity; and the subscripts $p$ and $s$ specify the ponded water and its stream replacement, respectively.

To quantify whether a ponded water or stream dominates nitrogen removal, and the hydrologic and biological tradeoffs, we examined the ratio of Damköhler numbers:

$$\frac{Da_p}{Da_s} = \frac{A_p}{L \cdot b} \cdot \frac{\nu_p}{\nu_s} \qquad (3)$$

where the first factor is the ratio of hydraulic loads $\left(\frac{A_p}{L \cdot b}\right)$, which we refer to as the hydrologic dominance index (HDI), and the second factor $\left(\frac{\nu_p}{\nu_s}\right)$ is the ratio of biological activities. Equation (3) shows that the threshold where a ponded water dominates nitrogen removal over its stream replacement occurs at HDI $= \frac{\nu_s}{\nu_p}$, which is where $\frac{Da_p}{Da_s} = 1$ and the proportion of nitrogen removed is equivalent between the ponded water and stream. As the HDI

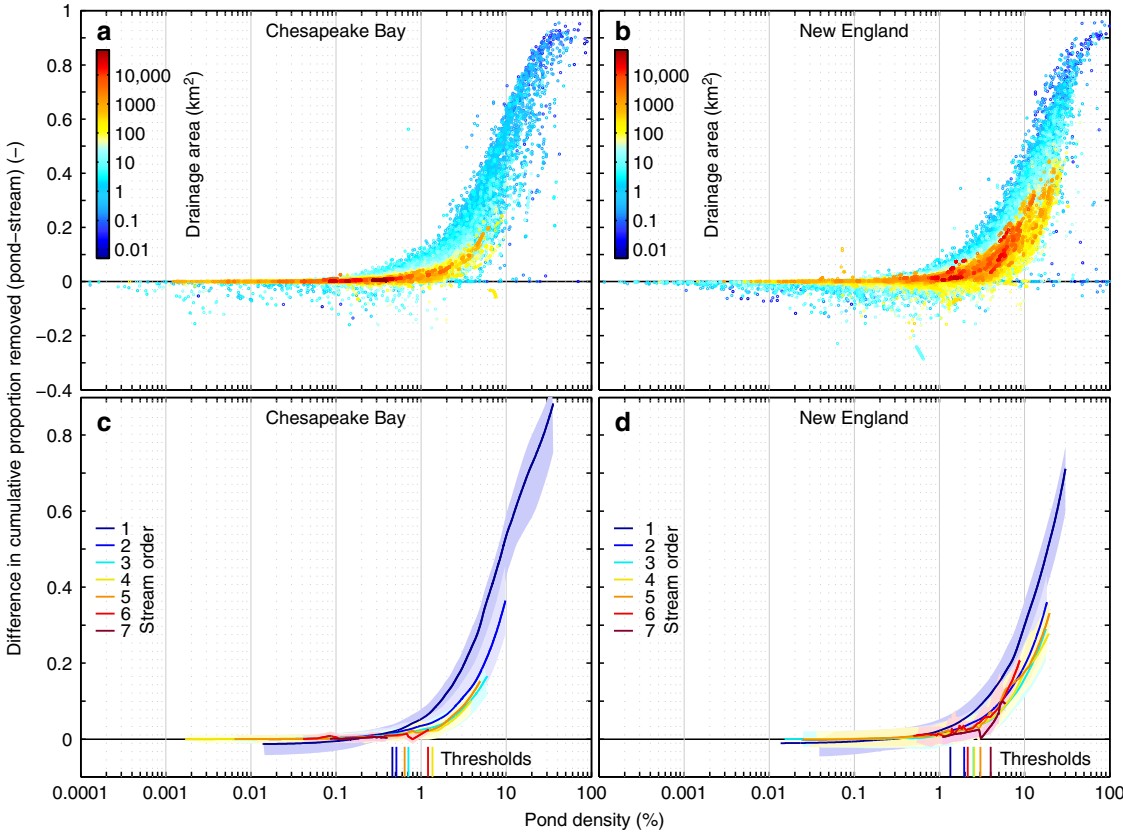

**Fig. 2** Cumulative proportion of total nitrogen removed annually by ponded waters relative to streams (expressed as a difference) for varying pond densities and network locations. Dot-plots are colored by cumulative upstream drainage area for the **a** Chesapeake Bay and **b** New England sub-regions (see Fig. 1). A locally weighted scatterplot smoothing was applied to the values per stream order to identify pond density thresholds for the **c** Chesapeake Bay and **d** New England sub-regions (see Methods). Shaded regions are the upper and lower limits of the scatterplot smoothing

increases, the dominance of the ponded water over its stream replacement increases (Fig. 3). However, Fig. 3 shows that there is a large range of nitrogen removal for a similar HDI, particularly at high values, revealing that there are other factors influencing hydrologic dominance, and in turn, nitrogen removal.

The degree of pond connectivity to the river network clearly could be a control on the variation in nitrogen removal (Fig. 3). We quantified pond connectivity as a scaled measure of pond centeredness on the river network:

$$\text{Pond connectivity} = \frac{L_{\text{centroid}}}{L} \qquad (4)$$

where $L_{\text{centroid}}$ is the distance from the planar centroid of the ponded water to the midpoint of the intersecting stream reach (m) (see Methods). This metric provides a measure of pond symmetry relative to the stream, where lower values indicate greater centeredness of the ponded water on the river network, suggesting from process-based studies[4,27,28] that the ponded water is more likely to be well mixed and better at removing nitrogen in contrast to one that is largely bypassed by flowing water. For example, consider two different ponded waters that have a similar HDI in relation to their stream replacements: the ponded water that is well connected to the network will remove more nitrogen than the one that is poorly connected (see colored dot-plot in Fig. 3). We found that the connectivity threshold (see Methods), where nitrogen removal by a ponded water relative to its stream replacement substantially decreases, occurs at a pond connectivity value of 0.36 (±0.09) for NE (Fig. 3) and 0.19 (±0.07) for CB (Supplementary Fig. 3 and Supplementary Table 3).

Pond roundness also should be important in characterizing hydrologic dominance and mixing[28]. Ponded waters that are rounder in shape (see Eq. (5) in Methods) and well connected to their river network have greater capacity for nitrogen removal, whereas narrow ponded waters that are well connected by following the run of the river tend to be stream-dominant (see shape examples in Fig. 3 and Supplementary Fig. 4). The shape of a ponded water is clearly important, and together with pond connectivity, provided more explanation for nitrogen removal dominance in ponded waters versus streams; the influence of a round ponded water will decrease the farther its center deviates from the river network.

We found that pond location within the network (cumulative upstream drainage area), pond connectivity, and pond roundness explain nearly 70% of the variance in the pond-stream dominance of nitrogen removal in CB and 60% in NE (see Methods and Supplementary Table 4). For both CB and NE, drainage area explains a majority of the variability because it is negatively correlated with the HDI (Spearmen rank correlation $\rho = -0.66$, $p < 0.001$). Ponded waters have their largest effect in headwaters because corresponding HDI values are highest (Supplementary Fig. 5). Rivers become larger lower in the network, generally reducing the HDI and, therefore, the dominance of a ponded water. Pond connectivity and roundness in CB explain roughly 12% and 7% of the variability, respectively. In NE, pond connectivity is slightly more important than roundness, explaining roughly 12% of the variability compared to 4% for roundness. Less connected and rounder ponded waters tend to increase the HDI. The most important physical metric for assessing pond-stream dominance is location because it is the largest control

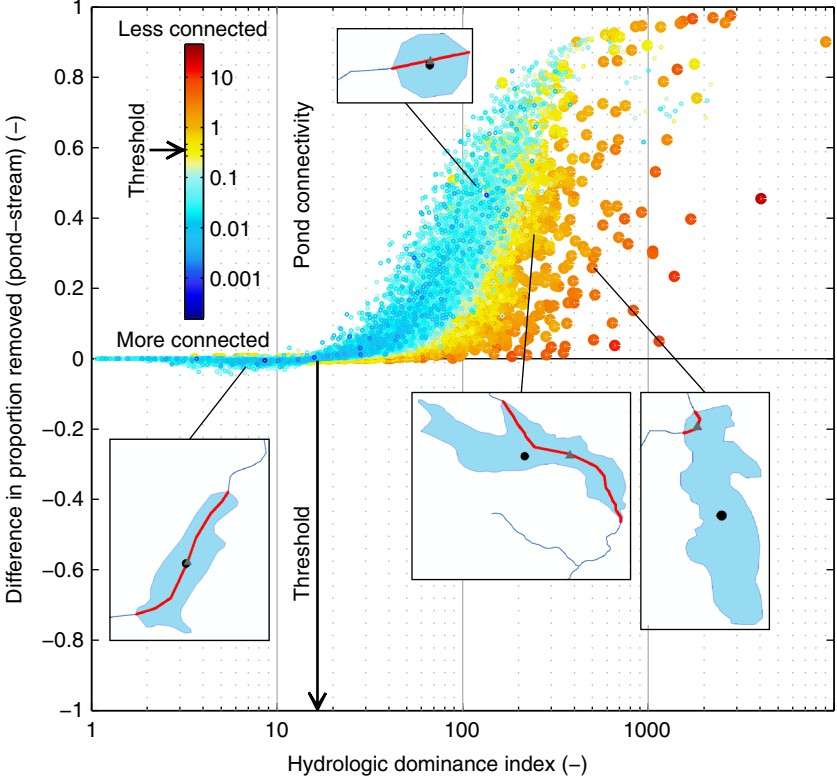

**Fig. 3** Proportion of total nitrogen removed annually by an individual ponded water relative to its stream replacement (expressed as a difference) across varying degrees of pond connectivity. The results shown here are of the New England sub-region and for ponded waters with only one intersecting stream reach (see Supplementary Fig. 3 for the remaining ponded waters and for the Chesapeake Bay sub-region). The dominance of a ponded water to remove nitrogen starts to decrease at a threshold in ponded connectivity of 0.36 (±0.09) (see Methods); the dot-plot is colored by pond connectivity (see Eq. (4)) where larger dots designate values above the threshold. Small values of pond connectivity indicate that the ponded water is more connected to the network while large values indicate the ponded water is less connected to the network, as illustrated by the shape examples. The black dots on the shape examples designate the centroid of the ponded water and the gray triangles designate the midpoint of the intersecting stream reach (in red). The threshold in the hydrologic dominance index where a ponded water becomes dominant occurs when it is equal to the ratio of stream to ponded water biological activities, which results from the case when the associated Damköhler numbers are equal (see Eq. (3))

on the HDI, followed by connectivity and shape that together significantly contribute to better explaining how ponded waters function relative to stream replacements.

In general, ponded waters remove more nitrogen than their stream replacements. However, there are some differences in efficiencies between sub-regions, with a smaller percentage of ponded waters being dominant in NE (77% dominant in NE versus 88% dominant in CB; Supplementary Table 1), which supports a conclusion that CB ponded waters are more efficient at removing nitrogen compared to NE ponded waters. Streams are up to 18 times more biologically efficient than ponded waters in NE and 14 times more efficient in CB (see Supplementary Table 5 and comparison to literature values in Supplementary Fig. 6). The result is a slightly larger threshold in HDI where ponded waters become dominant over streams in NE, with the underlying reason being that the threshold is shifted because the ratio of biological activities is marginally higher in NE.

**Cumulative outcomes in river networks**. This balance of hydrologic and biological factors affecting nitrogen removal at the local scale can be extended to provide a simple cumulative metric that allows for quick assessment of where and why ponded waters affect nitrogen removal throughout the entire river network, which will be important to evaluating the potential effectiveness of ponded water management on downstream nitrogen loading.

The cumulative HDI (see Methods) is a good indicator of how the entire river network is affected by replacing ponded

waters with streams (Figs. 1a, 4a) because this metric, unlike pond density, better reflects the balance of lotic and lentic blending. For example, ponded waters in the Appalachian Plateau in northern Pennsylvania more frequently dominate cumulative nitrogen removal (Fig. 1a) because of the generally larger HDI values in that area (Fig. 4a). These ponded waters also tend to be both more circular and well connected to the network, resulting in low cumulative pond connectivity values (see Methods) and an increased effect on cumulative nitrogen removal (Figs. 1a, 4b). Conversely, ponded waters in the New Jersey coastal plain are generally less efficient than their stream counterparts (Fig. 1a), which corresponds with low cumulative HDI and pond connectivity values that indicate narrow, well-centered ponded waters (Fig. 4a, b). Furthermore, less removal for a given pond density is due to a lower cumulative HDI, reflecting a greater degree of lotic blending and reduced effect of ponded waters.

In summary, our findings show that thresholds where ponded waters dominate cumulative nitrogen removal are suggested by pond density (Figs. 1, 2); however, pond density alone is not enough. In CB, pond density decreases with stream order, resulting in marginal coastal delivery reduction from ponded waters (Supplementary Figs. 1, 2). Conversely, pond density remains relatively constant in NE, corresponding with notable cumulative removal and reduced coastal delivery from ponded waters (~20% of total aquatic removal; Supplementary Figs. 1, 2). The underlying causes of reduced cumulative removal with increasing stream order for a similar pond density and differences

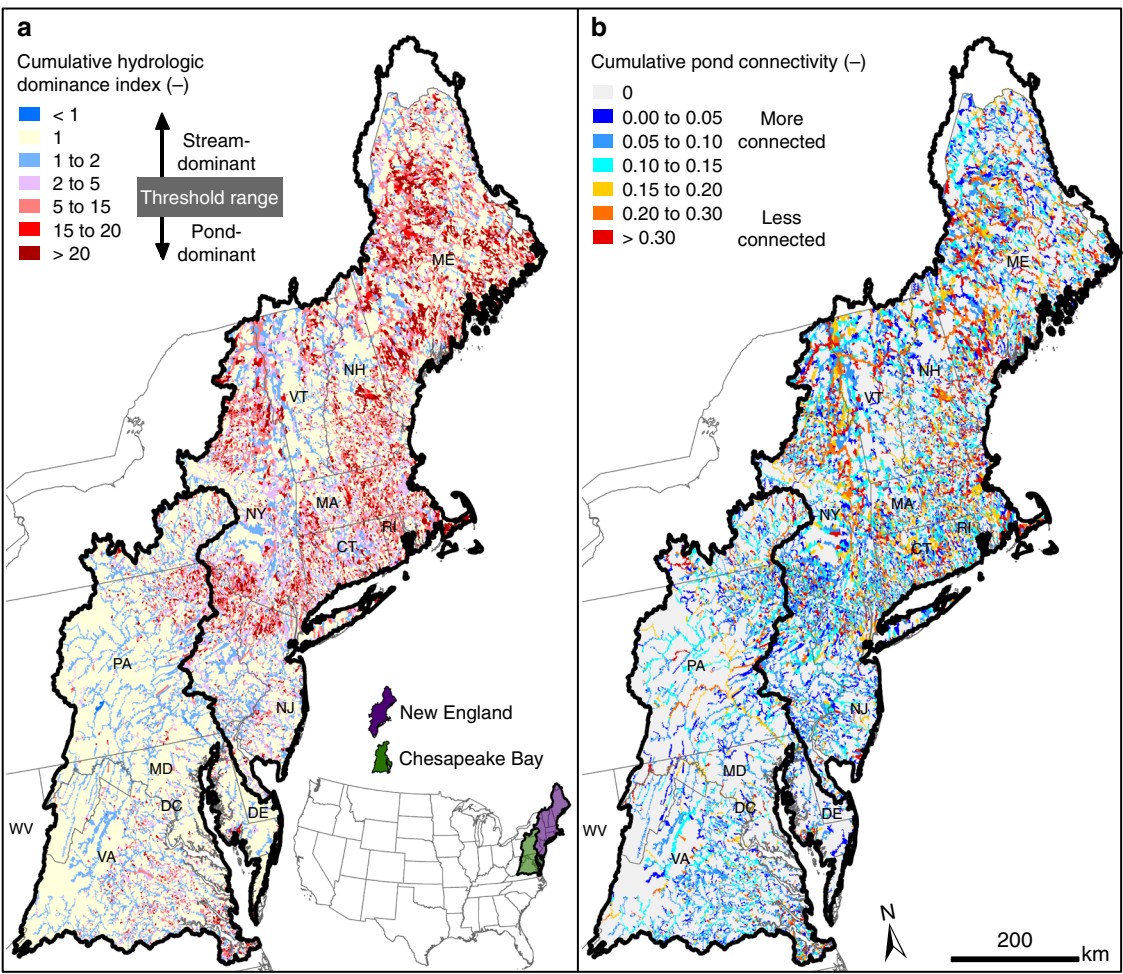

**Fig. 4** Hydrologic dominance of ponded waters as an indicator of nitrogen removal in the Northeastern United States. **a** The cumulative hydrologic dominance index throughout the river network (see Methods) indicates where nitrogen removal is dominated by ponded waters or by streams. Ponded waters are hydrologically dominant when the index is greater than one and streams are hydrologically dominant when the index is less than one. Locations void of ponded waters have an index value of one. Cumulative nitrogen removal is generally dominated by ponded waters in portions of the network at the threshold where this index is greater than the ratio of stream to ponded water biological activities. **b** Cumulative connectivity of ponded waters to the river network (see Methods). Smaller values indicate locations where upstream ponded waters are more connected with the river network. Larger values correspond with larger deviations from the network. Maps created using data from NHD[35] and expressions in the Methods

between sub-regions were only recognized by analyzing the balance of hydrologic and biological factors at the local scale, and additional metrics of pond shape and connectivity.

## Discussion

The dominance of a ponded water relative to a stream replacement depends on attributes of relative size, shape, and connectivity to the network. Previous studies have suggested that ponded water size (particularly surface area) and network location are important to the significance of nitrogen removal by ponded waters[1,21,22]. While in general agreement with our findings, these studies did not consider the balance between ponded water and stream hydrologic factors, or shape and connectivity characteristics. Consequently, those studies did not delve into regional, or within-network variation, or causes of thresholds. As a step towards explaining causation, an existing empirical relationship shows that nitrogen removal by individual ponded waters is strongly correlated with depth and residence time[1]; however, that relationship has been shown to be inconsistent between individual ponded waters[18]. In addition to

variation in biological activity, we illustrate that this relationship inconsistency is likely caused by pond shape, connectivity to the network, and position within the network (Figs. 2, 3, 4 and Supplementary Figs. 4, 5).

All of the metrics that were useful for explaining nitrogen removal by ponded waters (i.e., pond density, network position, HDI, roundness, and connectivity) are relevant to managing the water-quality functions of large river networks and identifying the tradeoffs associated with dam creation or dam removal at different positions in the network. While the number of large reservoirs is likely to increase worldwide[29], the more likely trend in the United States is dam removal due to age[30,31]. We showed that thresholds in pond density could be helpful in indicating how the management of reservoirs is likely to influence cumulative nitrogen loading. However, the pond density thresholds vary by location in the river network and therefore pond density alone is not a sufficient predictor (Fig. 2). The cumulative HDI better clarifies how location in the river network is important. For example, the lower cumulative HDI explains why ponded waters in the CB network only marginally decrease nitrogen loading compared to NE, which was not apparent from pond density.

The cumulative HDI is also potentially useful to quickly assess how management of ponded waters could reduce nitrogen loading to coastal areas. For the Northeastern United States, for example, a 1% increase in cumulative proportion nitrogen removed requires roughly a factor of two increase in the cumulative HDI. A possible management strategy in CB could be to increase the cumulative HDI with small ponded waters on headwater streams, beaver ponds for example, as these may be effective as restoration tools[16,32,33] by reducing downstream nitrogen loading. In addition to the HDI, we found that shape and connectivity factors are important.

Our findings suggest that if the reservoir has a circular shape and is well connected to the network, it will generally exceed the performance of a stream in the same location and lead to maximum removal of nitrogen. If the shape of the resulting reservoir is narrow and less circular, a stream could be more effective at removing nitrogen and, therefore, more effective at lowering nitrogen loading to downstream waters. Furthermore, a reservoir placed on a low-order stream will likely require more stringent shape and size scrutiny than on high-order rivers. A reservoir placed on a high-order river has a lesser effect on nitrogen removal because higher flows and the accompanying lower HDI outweigh the influence of reservoir shape (Supplementary Figs. 4, 5).

We conclude that interactions between the physical characteristics of ponded waters and the river network determine the individual and cumulative effects of ponded waters on downstream water quality and regional loading of nitrogen to coastal areas. As more physical information about ponded waters and stream channels becomes increasingly available throughout the nation and beyond, the effects of ponded waters can be readily predicted worldwide based on physical metrics, including position within river networks, pond density, and characteristics of pond size, shape, and connectivity to the network relative to a corresponding stream. Tradeoffs between the hydrologic factors affecting residence time and the biological activity in ponded waters determine their effectiveness relative to streams in removing nitrogen, leading to the identification of thresholds in densities, and size and shape characteristics, to evaluate when and where ponded waters significantly alter downstream water quality. As water resources are increasingly stressed worldwide, our findings are relevant to water managers who need practical tools to prioritize the creation of reservoirs or removal of dams, or the restoration of streams, to improve regional outcomes for river water quality and aquatic ecosystem health.

## Methods

**General approach.** We modified an existing spatially referenced model[24] to estimate biological activity in ponded waters and streams and annual total nitrogen loading throughout CB and NE using comprehensive river water quality, land use, and river network hydrography datasets. Stream channel and ponded water geometric properties were taken from the medium resolution (1:100,000) National Hydrography Dataset[34,35] (NHD Plus Version 2.1, https://nhd.usgs.gov/ accessed on May 6, 2017). From this dataset, we have length (m), mean annual discharge (m$^3$ d$^{-1}$), mean annual velocity (m d$^{-1}$), slope, stream order, catchment area (m$^2$), and cumulative upstream drainage area (m$^2$) for each stream reach (i.e., NHD flowline). For ponded waters, we have additional estimates of surface area (m$^2$) and depth (m). This dataset was used to populate the Spatially Referenced Regression On Watershed attributes (SPARROW) model[36] to estimate annual total nitrogen loading for every reach and ponded water in CB and NE (https://water.usgs.gov/nawqa/sparrow/). The model was further populated with land use and nitrogen source data consistent with ref. [24], but modified to upgrade from NHD Plus Version 1.1 to Version 2.1 to include the latest estimates of stream discharge and ponded water attributes (see details below). This model was calibrated to river water quality data to estimate biological activity in rivers, streams, and ponded waters for the two sub-regions and subsequently predict cumulative and local nitrogen loading throughout the river network for two cases: current conditions of the network including existing ponded waters and streams, and the same network with ponded waters replaced with estimates of streams. SPARROW fits uniform parameters over a large spatial domain and, therefore, provides parameters that can

be used to characterize the stream replacements. This characterization allowed us to perform a direct comparison without requiring an additional calibration (see details regarding model specifications, calibration, and nitrogen removal estimates below). Metrics of pond shape and connectivity were not brought directly into the SPARROW model; they were found important only by comparing the two cases.

To explain where, why, and how much ponded waters influence nitrogen removal relative to streams, we estimated the following local and cumulative metrics. At the local scale, hydrologic and biological factors controlling nitrogen removal were isolated by estimating the Damköhler number of each ponded water and stream replacement (see Eqs. (1) and (2)), assuming all benthic surface areas are equally reactive, flow is steady, and reaction processes are first-order. We used the uptake velocity as a measure of biological activity because it is a measure of uptake per unit area and, therefore, independent of hydrologic controls. The remaining physical parameters ($Q_p$, $Q_s$, $A_p$, $L$, $b$; Eqs. (1) and (2)) are provided by NHD and estimates of hydraulic geometry (see details below). For ponded waters that have several intersecting stream reaches (see details below), we estimated the HDI (Eq. (3)) for each unique ponded water by summing all the segmented ponded water surface areas and all the intersecting stream reach surface areas.

Roundness of each ponded water was evaluated with circularity (dimensionless):

$$\text{Circularity} = 4\pi \frac{A_p}{P^2} \quad (5)$$

where $P$ is the perimeter of the ponded water (m). Perimeter was estimated directly from the polygons provided by NHD. A circularity value of one indicates a near perfect circle and values approaching zero indicate an increasingly elongated ponded water. This shape metric is similar to the pond perimeter to surface area ratio used in other studies[37], but provides a more direct comparison to a circle.

In addition to local metrics, we estimated cumulative metrics to assess portions of the river network affected by ponded waters. Pond density was estimated as the cumulative surface area of ponded waters to cumulative drainage area:

$$\text{Pond density}_i = \frac{\sum_j^{N_i} A_{p,j}}{\text{DA}_i} \quad (6)$$

where $i$ is the reach index, DA is the cumulative upstream drainage area at reach $i$ (m$^2$), $N$ is the total number of ponded waters upstream of reach $i$, and $j = 1, \ldots, N$ is the ponded water index. Less than 1% of all pond density estimates were greater than one, which is due to error within some of the upstream NHD catchments that contribute to the summation of cumulative drainage area. We visually inspected each case and conclude that these catchments are incorrect. Therefore, pond densities greater than one were omitted.

The HDI used to describe hydrologic dominance between a ponded water and stream replacement at the local scale (see Eq. (3)) was applied to estimate a cumulative HDI throughout the entire river network as:

$$\text{Cumulative HDI}_i = \frac{\sum_1^{N_i} A_{\text{existing},i}}{Q_{\text{existing},i}} \frac{Q_{\text{replaced},i}}{\sum_1^{N_i} A_{\text{replaced},i}} = \frac{\sum_1^{N_i} A_{\text{existing},i}}{\sum_1^{N_i} A_{\text{replaced},i}} \quad (7)$$

where $A_{\text{existing}}$ is the wetted surface area of reach $i$ in the existing river network that is either a ponded water or a stream (m$^2$) (i.e., the summation includes all surface areas of existing ponded waters and streams upstream of reach $i$); $A_{\text{replaced}}$ is the wetted surface area of reach $i$ in the network where ponded waters were replaced with streams (m$^2$) (i.e., the summation includes all existing streams and stream replacements upstream of reach $i$); $Q_{\text{existing}}$ and $Q_{\text{replaced}}$ are the mean annual discharges of reach $i$ in the existing and pond-replaced river networks (m$^3$ d$^{-1}$), respectively, but assumed equivalent over the annual time period. The cumulative HDI represents the relative blending of all lotic and lentic waters upstream of a location in the river network. A standard metric to represent the hydraulics of a waterbody is the reciprocal hydraulic load (see Eqs. (1) and (2)). Based on standard engineering principles (e.g., ref. [38]), the hydraulic load itself is an intensive variable and not directly additive. Therefore, we accumulated the surface area independently and then divided by discharge, which are both extensive variables, to arrive at an estimate for the cumulative time it takes to displace all upstream water. By comparing the existing and pond-replaced river networks, the cumulative HDI indicates portions of the network that are either hydrologically dominated by ponded waters or streams. Locations in the network that are void of any upstream ponded waters (i.e., no stream replacements) have a cumulative HDI of one. Values greater than one indicate portions of the network that are hydrologically dominated by ponded waters and values less than one indicate portions of the network that are hydrologically dominated by streams. Portions of the network where ponded waters dominate nitrogen removal are, therefore, generally indicated by a cumulative HDI that is greater than the ratio of biological activities $\left(\frac{\nu_s}{\nu_p}\right)$.

Pond connectivity was estimated to assess the degree of centeredness of a ponded water on the river network. For each ponded water, we estimated the geographic coordinates of the planer centroid and the midpoint of the intersecting reach. With these coordinates, we calculated the distance from the centroid to the midpoint ($L_{\text{centroid}}$, m). For ponded waters that have several intersecting stream reaches (see details below), we estimated $L_{\text{centroid}}/L$ (Eq. (4)) for each segment.

Then to provide a single value for each unique ponded water, we estimated the pond connectivity value as the mean of all corresponding pond segments. The average tendency of the connectivity of ponded waters throughout the river network (i.e., the average connectivity of all ponded waters upstream of a point in the network) was estimated as:

$$\text{Cumulative pond connectivity}_i = \frac{1}{N_i}\sum_j^{N_i}\left(\frac{L_{\text{centroid},j}}{L_j}\right)_i \tag{8}$$

**River network hydrography.** We accounted for only ponded waters connected to the river network (i.e., has a corresponding stream reach) that are either a natural lake or reservoir. In total, we used 191,256 stream reaches with corresponding catchments (84,214 in CB and 107,042 in NE), of which 29,046 are covered by ponded waters (15%). The river network consists of 289,214 km (128,243 km in CB and 160,971 km in NE), with 6.4% of the total length covered by ponded waters (Supplementary Table 1). We excluded reaches that are geographically isolated, are diversions or canals, or are directly along coasts.

Some larger ponded waters have several intersecting stream reaches, which results in segmentation of a ponded water. There are a total 18,180 unique ponded waters (4305 in CB and 13,875 in NE), of which 2804 are segmented (generally the larger lakes and reservoirs), resulting in a combined total of 29,046 unique ponded waters and segments (6225 in CB and 22,821 in NE). NHD provides estimates of surface area ($m^2$) and mean annual discharge ($m^3\ d^{-1}$) for each pond segment, which allowed us to quantify different proportions of nitrogen removed for each segment. We chose to account for ponded water segmentation to provide better estimates of nitrogen removal because different portions of a large lake or reservoir have been shown to vary in removal efficiencies[4].

**Ponded water replaced by a stream.** Following model calibration (discussed below), we repopulated the SPARROW model with river network hydrography in the absence of ponded waters. To replace a ponded water with a stream reach, we used estimates of stream travel time (d), stream depth (m), stream reach length (m), velocity (m $d^{-1}$), and wetted width (m). We used the Jobson Method[39] and ref. [40] to estimate velocity, depth, and travel time for the new stream reach replacements, in addition to all stream reaches. The Jobson Method uses drainage area, stream discharge, reach length, and slope. Because velocity (and travel time) is sensitive to slope, we used updated slopes (enhanced NHD Plus Version 2.1). We assume that the slopes of the ponded water reaches are representative of the steepest, most direct path through the network and representative of a stream. Stream depth was estimated using $d = 0.261Q^{0.3966}$ (m) with $Q$ in units of $m^3\ s^{-1}$, which is based on a hydraulic geometry study of 112 river locations in the United States[19,40]. With estimates of velocity, depth, and stream discharge, we estimated wetted width. We assume that discharge through a ponded water and its stream replacement are consistent over the annual time period. We interpret the stream reaches intersecting ponded waters (i.e., artificial NHD flowlines) as indicative of true stream geometry and routing. These flowlines may be artificially short due to lower than natural sinuosity (approaching one), which would reduce the estimated stream travel time and, therefore, lessen nitrogen removal by the stream replacement. As a test, increasing the sinuosity of these flowlines to two for a typical meander results in a stream travel time increase by a maximum factor of two, assuming velocity remains unchanged. Because ponded waters generally have reciprocal hydraulic loads much larger than their stream replacements (on average 7.5 times larger), doubling the stream travel times does not change the overall patterns of results or conclusions. Furthermore, we anticipate that minor adjustments to sinuosity have negligible influence on pond connectivity estimates because artificial flowlines represent the most direct downstream path.

**SPARROW model outputs and proportion removed.** The SPARROW model produces estimates of annual total nitrogen loading (incremental and cumulative) entering and exiting each stream reach and ponded water. SPARROW maintains the assumptions that all benthic surface areas are equally reactive, flow is steady, and reaction processes are first-order[36].

The incremental efficiency of total nitrogen removal in the streams was estimated within SPARROW as:

$$\frac{\text{Load}_{\text{OUT},s,i}}{\text{Load}_{\text{IN},s,i}} = \exp\left(-\nu_s \frac{\tau_i}{d_i}\right) \tag{9}$$

where $\text{Load}_{\text{IN},s}$ is the total nitrogen load entering reach $i$ which includes load entering the upstream end of the reach and load generated within the corresponding catchment (kg $y^{-1}$), $\text{Load}_{\text{OUT},s}$ is the total nitrogen load exiting reach $i$ after aquatic decay (kg $y^{-1}$), $\tau$ is the mean annual stream travel time (d), $d$ is the mean annual stream depth (m), and $\frac{\tau}{d} = \frac{L \cdot b}{Q_s}$ (see Eq. (2)). Load entering the upstream end of the reach experiences the full travel time while load generated within the catchment is assumed to enter at the midpoint of the reach and,

therefore, experiences half of the travel time. This efficiency (Eq. (9)) was estimated for each existing stream reach and stream reach replacement.

The incremental efficiency of total nitrogen removal in ponded waters was estimated within SPARROW as:

$$\frac{\text{Load}_{\text{OUT},p,j}}{\text{Load}_{\text{IN},p,j}} = \exp\left(-\nu_p \frac{A_{p,j}}{Q_{p,j}}\right) \tag{10}$$

where $\text{Load}_{\text{IN},p}$ is the total nitrogen load entering ponded water $j$ which includes load flowing into the upstream end of the ponded water and load generated within the corresponding catchment (kg $y^{-1}$), and $\text{Load}_{\text{OUT},p}$ is the total nitrogen load exiting ponded water $j$ after aquatic decay (kg $y^{-1}$). Each unique ponded water (or ponded water segment) is assumed to be completely mixed. Therefore, both the load entering the ponded water and load generated within the catchment experience the same travel time.

For each ponded water and stream replacement, we estimated the proportion of total nitrogen removed as a basis for comparison ($R_p$ and $R_s$, respectively):

$$R_{p,j} = 1 - \frac{\text{Load}_{\text{OUT},p,j}}{\text{Load}_{\text{IN},p,j}} \tag{11}$$

$$R_{s,j} = 1 - \frac{\text{Load}_{\text{OUT},s,j}}{\text{Load}_{\text{IN},s,j}} \tag{12}$$

For a segmented ponded water, a single value of $R_p$ was computed as the proportion removed by the entire ponded water and a single value of $R_s$ was computed as the combined proportion removed by all of the interesting stream reaches. To compare nitrogen removal by a ponded water relative to its stream replacement, we estimate the difference in proportion removed as:

$$\Delta R_j = R_{p,j} - R_{s,j} \tag{13}$$

where a positive difference indicates nitrogen removal is dominated by the ponded water and a negative value indicates that nitrogen removal is dominated by the stream. We chose to use a difference because it provides a direct relative measure of efficiencies, rather than use another metric such as a ratio that could provide misleadingly large numbers (e.g., a ponded water removed 10% while a stream removed 0.01%, resulting in a ratio of 1000).

We estimated the cumulative proportion of total nitrogen removed throughout the entire river network for two cases: existing river network including ponded waters ($R_{\text{accum},p}$) and the same river network with ponded waters replaced with streams ($R_{\text{accum},s}$):

$$R_{\text{accum},p,i} = \left(1 - \frac{\text{Load}_{\text{accumOUT},p,i}}{\text{Load}_{\text{NODECAY},i}}\right) \tag{14}$$

$$R_{\text{accum},s,i} = \left(1 - \frac{\text{Load}_{\text{accumOUT},s,i}}{\text{Load}_{\text{NODECAY},i}}\right) \tag{15}$$

where $\text{Load}_{\text{NODECAY}}$ is the cumulative total nitrogen load exiting reach $i$ without experiencing aquatic decay (i.e., the total cumulative nitrogen load entering the river network from the landscape), which is equivalent between the two cases (kg $y^{-1}$). $\text{Load}_{\text{accumOUT},p}$ and $\text{Load}_{\text{accumOUT},s}$ are the estimated loads exiting reach $i$ for the two cases, respectively, which represent an accumulation of all decay processes upstream. To compare cumulative proportion of nitrogen removed by a river network with ponded waters relative to a river network of only streams, we estimated the difference in cumulative proportion removed as:

$$\Delta R_{\text{accum},i} = R_{\text{accum},p,i} - R_{\text{accum},s,i} \tag{16}$$

which provides a cumulative estimate of the role of ponded waters relative to streams throughout the entire river network (see Figs. 1a, 2).

**Threshold estimation and statistical analysis.** Through the comparison of an existing river network to one in which all ponded waters were replaced with streams, we identified thresholds in readily available physical metrics where ponded waters become important features of the aquatic landscape. We identified three types of thresholds: a threshold in pond density, a threshold in the HDI, and a threshold in pond connectivity.

A threshold in ponded density where the cumulative role in ponded waters becomes important was quantified using a non-parametric regression. A locally weighted scatterplot smoothing was applied to 98% of the values (to reduce the influence of extreme values) per stream order. The upper and lower limits of the regression were estimated by applying the same non-parametric regression to the positive (upper limit) and negative (lower limit) residuals. We define the threshold in pond density as the point where the lower limit is consistently

above zero at every increasing pond density value (Fig. 2 and Supplementary Table 2).

At the local scale, Eq. (3) shows that the threshold in the HDI occurs when it is equal to the ratio of biological activities $\left(\frac{A_p}{L \cdot b} = \frac{v_s}{v_p}\right)$. A threshold in pond connectivity (Eq. (4)) was quantified where a significant break in slope between pond connectivity and the difference in proportion removed occurs (using the Segmented R package[41,42]). We applied this breakpoint analysis up to the 99th percentile of pond connectivity values for CB and NE, providing the estimate of the threshold with 95% confidence intervals (Supplementary Table 3).

One of our main conclusions is that the role of an individual ponded water relative to its stream replacement is controlled by network location, the degree of centeredness on the network (pond connectivity), and shape (circularity). We applied a multilinear model to evaluate percent variance explained and relative importance of each physical metric to assessing how a ponded water functions differently than a stream:

$$\ln\left(R_p\right) - \ln(R_s) = \beta_0 + \beta_1 \ln(c) + \beta_2 \ln(s) + \beta_3 \ln(\text{DA}) + \varepsilon \qquad (17)$$

where $c$ is pond connectivity, $s$ is circularity, the $\beta$s are the regression coefficients, and $\varepsilon$ is random error. We applied an analysis of variance to the regression model to quantify the sum of squares of each coefficient and estimate the percent variance explained and relative importance by each metric (Supplementary Table 4). In addition, we applied an analysis of variance to nested models to test the explanatory value of including connectivity and roundness separately, for which results were consistent with the full model (Eq. (17)) and, therefore, not reported.

**SPARROW model specifications and calibration.** We used the same sources and land-to-water delivery variables as ref. [24] to provide aquatic decay estimates and mean annual total nitrogen loads (Supplementary Table 5). The nitrogen sources accounted for in the SPARROW model are point sources generated from wastewater treatment facilities and sources generated within each catchment that include: atmospheric deposition; fertilizer and fixation for corn, soybeans, and alfalfa crops; crops other than corn, alfalfa, and soybeans; manure from livestock; and developed land. These sources represent land use conditions as of 2002[24,43], which are the latest published estimates. Nitrogen data used were retrieved by refs. [24] and [43] primarily from the USGS National Water Information System and the US Environmental Protection Agency's Storage and Retrieval database. However, to use the latest estimated network hydrography, ponded water size, and mean annual flows, this existing nitrogen dataset was updated from NHDPlus Version 1.1 (V1) to Version 2.1 (V2). Because we are investigating the influence of ponded waters on nitrogen removal and not on current conditions of nitrogen sources or water quality, this upgrade was necessary. To upgrade from V1 to V2, the intensive variables (land-to-water) and point sources for each reach remain unchanged (i.e., copied straight across the two versions using the NHD crosswalk). The extensive variables (land sources) were transferred from V1 to V2 by normalizing all sources by the corresponding V1 catchment incremental area, assuming sources are spread uniformly over each catchment, and copied to V2. There were some V2 reaches that did not have corresponding V1 reaches. Therefore, we applied a fill routine to maximize that amount of spatially explicit information. This fill routine linearly interpolated area-normalized sources between adjacent V2 catchments. These normalized sources (and those filled) were multiplied by the V2 catchment area to complete the upgrade.

While the annual total nitrogen SPARROW model is similar to ref. [24], it was recalibrated with the updated NHDPlus V2 attributes and specified to provide region-specific biological activity estimates (uptake velocities of streams, rivers, and ponded waters (m d$^{-1}$)) for CB and NE. We specify unique regions because streams and reservoirs have been found to behave differently in terms of nitrogen decay in CB relative to NE[44]. Consistent with specifications by ref.[24], we estimated different uptake velocities for small streams (mean annual discharge $\leq 2.83$ m$^3$ s$^{-1}$) and large rivers (mean annual discharge $> 2.83$ m$^3$ s$^{-1}$) for CB and NE. Small streams are statistically significant to the predicted nitrogen removal while large rivers are not (Supplementary Table 5). Therefore, to achieve the most likely estimates for nitrogen uptake in streams, we make this specification for two stream size classes. Our goal was not to create a new SPARROW model or loading results, but rather estimate biologic activity based on the best available land use and water quality data to isolate the cumulative effects of ponded waters and explain regional trends with readily available physical metrics.

The nitrogen sources applied in the model are spatially variable and based on comprehensive datasets[24]. Spatial variability in sourcing will affect the amount of nitrogen mass removed[22]. As a check, we re-predicted nitrogen loads with spatially uniform sources, keeping the calibrated aquatic decay estimates consistent. With these predictions, we repeated the analysis by re-computing the cumulative and individual proportions of nitrogen removed by ponded waters and streams. Because we are representing the efficiency of ponded waters and streams by proportion removed (Eqs. (14) and (15)), applying spatially uniform sources did not change our results. Therefore, all the results presented here are based on predictions made using spatially variable nitrogen sources.

**Data availability.** We used the publicly available National Hydrography Dataset (https://nhd.usgs.gov/). The SPARROW model is also publicly available (https://water.usgs.gov/nawqa/sparrow/). The SPARROW data input file and code specifications are based on previously published work. Expressions and equations in the main text and Methods sections can be used to reproduce the analyses and results.

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

## Acknowledgements

The work is a product of the John Wesley Powell Center River Corridor Working Group, supported by U.S. Geological Survey and National Science Foundation Hydrologic Sciences Program. Any use of trade, firm, or product names is for descriptive purposes only and does not imply endorsement by the U.S. Government. Any opinions, findings, and conclusions or recommendations expressed in this material are those of the authors and do not necessarily reflect the views or policies of the USGS.

## Author contributions

N.M.S. and J.W.H. conceived the study and wrote the paper with contributions from R.B.A., G.E.S., R.B.M., K.E., J.D.G.V., E.W.B., and D.S. The data compilation and modeling were performed by N.M.S., J.W.H., R.B.A., G.E.S., and R.B.M. All authors interpreted results.

## Additional information

**Competing interests:** The authors declare no competing interests.

