## [Peer Review File · Nature Communications]

Reviewers' comments:

Reviewer #1 (Remarks to the Author):

In this paper, the authors attempt to quantify the degree to which ponds or streams dominate nitrogen removal from river networks at regional scales. The authors also use a number of landscape metrics to explain the degree to which streams or ponds dominate nitrogen removal. Nitrogen removal in both streams and ponds is estimated using the popular SPARROW model. Landscape metrics are estimated using spatial data and GIS techniques.

The authors find that the pond density, stream order, drainage area, and degree of connectivity all influence the degree to which nitrogen removal is influenced by ponds and streams. The authors also provide a novel metric called the hydrological dominance index. This index is based on theory borrowed from chemical engineering and applied to environmental chemistry. This novel metric also explains whether nitrogen removal is dominated by streams or ponds.

Overall I found the study to be well designed and the methods appropriate. The findings are quite novel and interesting, and deserve to be seen by hydrologists everywhere. The threshold relationship between some of the metrics and pond or stream dominance of nitrogen removal is particularly interesting and novel. However, I do think that a little more is needed before this paper deserves to be in a Nature journal. I will lay out my arguments below and the editors may make the final decision.

The main shortcoming is that a large number of landscape factors, some novel, are applied to explain variability of pond vs stream dominance of nitrogen removal. The plots are compelling and beautiful. However, I wanted a quantitative basis for many of the statements the authors make. First, they find a number of thresholds in the relationships between variables. Yet no threshold is quantified. Quantifying a threshold is a fairly straightforward statistical operation that should be done here. Second, the authors leave the reader with the sense that there are a lot of metrics that work. But which one, if any, is better? Surely a quantitative comparison is warranted here. Why not calibrate a simple statistical model to estimate how much of the variability in pond vs. stream dominance of nitrogen removal is explained by each factor? It should also be possible and straightforward to assemble multi-variate models that explain the proportion of variance explained by combinations of factors. Indeed, the authors actually make a statement to the effect that the factors in Figure 3 explain more variance than the factors in Figure 2, yet no variances are quantified anywhere. In my mind, this lack of statistical analysis falls below the standard of due diligence of Nature.

I also thought that there have been a few papers recently published on similar topics that could be referenced here. They are:

Biogeochemical hotspots: Role of small water bodies in landscape nutrient processing

FY Cheng, NB Basu

Water Resources Research

Do geographically isolated wetlands influence landscape functions?

MJ Cohen, IF Creed, L Alexander, NB Basu, AJK Calhoun, C Craft, ...

Proceedings of the National Academy of Sciences 113 (8), 1978-1986

Geographically isolated wetlands are important biogeochemical reactors on the landscape

JM Marton, IF Creed, DB Lewis, CR Lane, NB Basu, MJ Cohen, CB Craft

BioScience 65 (4), 408-418

A better situation in the existing literature would better establish the need for this study.

If these issues are resolved, and they should be fairly straightforward to resolve, then I believe the paper would be publishable in Nature Communications. Thank you for giving me the opportunity to review such an interesting paper!

Reviewer #2 (Remarks to the Author):

This is an interesting and well-written study that shows that when ponded waters replace channelized waters, nitrogen removal tends to increase, but that there is variability in the increase caused by pond characteristics. This result was obtained using the SPARROW model calibrated to the Chesapeake Bay and New England regions. The approach was to parameterize SPARROW with ponds based on existing spatial data, and then in a scenario, replace ponded waters with their underlying channels. The study also looks at various ponded water attributes, including their position in the network, degree of connectivity, and size relative to drainage area, identifying these as a source of considerable variability in N removal, beyond just the ponds hydraulic load. The major take home message is that ponded waters, and their geometry, should be considered when developing management strategies to combat nutrient pollution.

Conceptual Comments

The analysis, results, and presentation seem mostly sound (with some technical questions below). However, the finding that ponded waters tend to have more removal than streams they replace does not seem particularly novel. I was surprised that the network scale role of ponded waters (as opposed to local role of ponded waters replacing local stream channels) was not emphasized more. It seems like the more important points of the analysis are not really emphasized - really only mentioned briefly at one point in the main text. The results on cumulative role of ponded waters at scale of the entire region show that ponded waters as a whole, contribute very little to aquatic N removal compared to channelized streams and rivers (Table S1). Removal by the network with streams only was 47.4% and 22.2%, and with all ponds added, removal increases by <2%. In the scenario with ponds, they represent a slightly larger component of removal by the entire network (15.7% and 4.5%), but that must be because removal by ponds simply replaces removal by streams (either locally, or because downstream rivers could remove excess from upstream if ponds are removed). This is very interesting and useful result and should be emphasized and explained further. If it is the true that rivers dominate network scale removal, then why is that the case? Their biological activity as uptake velocity (Table S2) is about 10x higher than in ponds (at least in low orders). This is also an interesting result that is not really emphasized at all. It would be helpful to also report the total surface area of streams and rivers in each region (with and without ponds). The fact that stream reactivity appears to be 10x high in streams and rivers compared to ponds (which is counter to some examples reported in the literature) and that as a result the cumulative ponded role at network scale is small, seems to me to be more relevant regional result that SPARROW is ideally suited to demonstrate.

The current emphasis on what happens if ponded areas replace streams is still useful, but if ponds as a whole are not a big factor, will managing them (e.g. via reservoirs as discussed later in the text) really make a difference? Any one reservoir seems it would have little impact, in larger rivers because dominance index is small and not affected by shape, and in headwaters because you would need very many of them as any one will intercept little N load. This has implications for which kinds of dam removals to emphasize, if you are factoring in their role in N removal in whether to take down.

Technical Comments

I could not figure out how pond characteristics other than uptake velocity and pond surface area could be assessed in the SPARROW model as described in the text. The importance of pond characteristics is a major point of the paper. Pond connectivity in particular was identified as an important secondary factor after hydraulic load. The metric for connectivity indicates that ponds that are little connected have less removal (as in bottom right inset in figure 3). But if that pond in Figure 3 is replaced by a stream, then the stream is very short ($L \cdot b$ is small compared A of the pond). The SPARROW model assumes all ponded surface areas have the same reactivity (L354)

and all ponds assumed to be completely mixed (L372). The model is calibrated with only v_f and HL in the removal equations. So how does connectivity factor in? I may be missing something very simple, but I think this should be addressed clearly and explicitly in the methods to increase understandability for most readers.

I was also confused by some of the terminology. Does the term "cumulative nitrogen removal" (L96, L184) refer to total nitrogen removal within an NHD subcatchment (I think it does based on the figures 1b and 2), but I would normally interpret cumulative as meaning total removal by all lakes in the upstream catchment. This term should be clearly defined in this context. Is this term the same as "cumulative effects" in larger river basins (L101)? Is this meant to be cumulative effects in all large rivers (which is sum of local) or entire river basin (which is sum of all upstream). I would review the use of these terms to make sure they are clear.

Another was the term used for the aquatic decay parameters in Table S2. I think these should be reporting uptake velocity (as discussed in the main text), but in S2 the "inverse hydraulic load" is reported. L373 points out the reciprocal hydraulic load, but this is not what is in Table S2.

Hydraulic load is not the decay coefficient and I don't think is calibrated, but is instead specified based on channel characteristics (discharge, width, length). I think this is really biological reaction time scale? But I would report as uptake velocity, as it is more intuitive and reported in field studies. Comparison with empirical measurements of this result, as done one of the early Alexander et al. 2000 papers would be helpful to place these modeling results in appropriate context.

Specific Comments

L56. Say depth and residence time. (Volume to surface area ratio = depth)

L84-85. Is creating more dams likely? It would probably be worth mentioning the trend of dam removals due to age occurring in the region, and debates on whether to replace.

L105-106. Can you be more explicit on how it explains?

L122. I think that it would be helpful to note that the hydrologic ratio will always be greater than one, but that the size of the ratio determines the hydrologic dominance.

L147. In text, threshold is 1, but in figure, it is 0 (log of 1). I would make consistent.

L157. What is meant by cumulative effects? Within NHD subbasin or in entire upstream watershed, or in entire region? I suggest being more specific.

L161. Tie this back to total removal by lakes in entire river network. Even though NE has fewer ponds that are dominant, because there are more of them (or more total surface area), ponds play a more important role in NE than in CB.

L168. This difference in biological activity between streams and ponds should be emphasized much more, including relating to empirical measurements in the literature. Is this consistent with empirical observations? Rivers and ponds are less different (if you average NE and CB). The difference between ponds and small streams is much greater than difference between ponds in CB and NE (only 2x different).

L184. This should clearly state that ponds dominate over their stream replacement only, not at network scale (statement could be misinterpreted as written).

L197. But if ponded waters have such a small role at the scale of entire network, will management of individual ponded water bodies really be effective?

L205. Improve water quality for N, but maybe not for other things (use of DO, pathogens, etc).

L261. Is pond density for the entire upstream drainage area (which I would call cumulative) or just for the NHD subcatchment. Its described slightly differently in different places. L265 suggests it is for subcatchment. Figure S1d suggests it is per stream order. Please clarify.

L284. This index is only about those stream reaches that have ponds, and not all stream reaches, right? Because it is stated that the index is applied to the entire network, it could be interpreted as the relative importance of ponds vs. all streams. Based on results in Table S2, ponds are relatively unimportant. I would clarify in this section that this is the case. Also, L171 is a bit misleading because it says that this index is a good indicator of pond dominant parts of the network. This has not been demonstrated because it only talks about those reaches that have ponds and not all the connecting reaches that have no ponds.

L293. Explain purpose of this analysis in opening sentence (i.e. estimate connectivity).

L306. Is perimeter of the ponded water in the NHD plus? I don't think this attribute was stated when discussing the data set.

L390-399. Confusing. How is equation sensitive to circularity or connectivity?

L407-408. Variable abbreviation identical.

Figures. The vertical lines in Figure 3 and S 2-4 show the v ratio where removal by ponds is greater than streams. It took me a while to understand what the vertical lines represent and should be explained more clearly. I think an arrow pointing at the X axis value of hydrologic ratio could be better. These figures captions should define that the ratio of biological activity has same units hydrologic index, and is the value of hydro index when role of ponds and streams are equal. Figure 1B - show scale values not in log.

Figure 3. Mismatch in use of pond density in terms of % in text and log proportion in figure. Make consistent.

EQN 6 - is this only of reaches with ponds? If so, its not really an index of the whole watershed relative importance of ponds to stream reaches. I would rather see an index of the whole network, not just the reaches with ponds.

EQN 9 and 10. Why use different equations? Why not use exponential form for ponded? Seems more parsimonius to use the same equation.

Supplemental Figures 2-4. Alter titles. All main titles are identical and it takes a while to figure out what is different about each one.

Reviewer #3 (Remarks to the Author):

The manuscript entitled "Thresholds of lake and reservoir connectivity in river networks control nitrogen removal" focusses on effect of different characteristics of ponded waters on nitrogen removal. What is particularly new, is that ponded waters are compared with their stream replacements. Such comparison is of great value to managers that want to construct or remove reservoirs e.g. for clean drinking water provisioning. The main conclusion is that pond connectivity and hydrologic dominance index have most effect on nitrogen retention.

The comparison between hydrological and biological factors is intriguing. At the same time the current introduction of these factors makes things complicated to understand (see my point P1). Furthermore I had some other comments and suggestions on the definitions (P2) and conclusions (P3). After my main points I have some other suggestions and comments.

Main points:

P1. The hydrologic dominance index is a nice factor to explain the effect of hydrology. However, it took me a while to understand the vertical line representing the biological factors as described by $v_s:v_p$ in most of the figures. After rewriting several equations I found the logic behind it (at least I think so). The understanding of the $v_s:v_p$ is important to understand the thresholds that are described in the paper. I would therefore suggest that the authors clarify the manuscript and figures with respect to the thresholds and the equations, preferably already in the main text (before the methods).

P2. I had difficulties with understanding the term 'threshold'. After reading the title it was not yet clear to me what kind of thresholds the authors meant. As connectivity was mentioned in the title I was thinking in the direction of hydrological connectivity thresholds. Then two new terms in the abstract ("thresholds in pond density" and "cumulative thresholds") did confuse me as I missed the term "connectivity" in relation to "threshold". After reading the manuscript I had the feeling there where many thresholds, in size, shape, connectivity, position and density, some of them being more important than others. I would suggest that the authors would be more clear in the title, abstract and introduction about what kind of threshold(s) they are looking at.

P3. The conclusions of the authors are that the different factors (e.g. size, shape, connectivity, position and density) are not equally important in the effect on nutrient retention. (see for example line 153 onwards: "While the shape of a ponded water is clearly important... water versus streams". How is it tested that the shape of a ponded water has less effect on nitrogen removal than pond connectivity and hydrologic dominance index? I missed the statistical explanation on

which this statement is based.

Other points:

P4. What are the main pitfalls of the used modelling approach including the stream replacements? How do they influence the results?

P5. Line 36: what is meant by accumulated biogeochemical processes?

P6. Line 42 and onwards: According to the authors high nutrient retention in constructed reservoirs is an issue because of the potential of cyanobacterial blooms and related toxins, while this is seems not been recognized as an issue in a constructed or natural pond. It is not clear to me how this differences come about.

P7. Line 48: what is meant by biogeochemical hotspots?

P8. Line 53: 'Although' suggest a contradiction, however I don't understand the contradiction. The first part of the sentence seems to contain the same information as the second part.

P9. Line 104 onwards: "In addition to ... two regions." I don't understand this.

P10. Equation 1: if τ (tau) is the mean annual stream travel time I would think that $A*d/Q$ is the mean annual pond travel time (or better 'residence time'). However, d seems to be the depth of the stream, and not the average depth of the lake. Could the authors clarify this more?

P11. Additionally at equation 1, the unit of $L_b (=d[m]/(Q[m^3/d]*\tau[d]))$ is meters according to the explanation, however a simple unit analysis shows that this should be m^2 , but then it could not be the length of the stream as stated in line 118. Reading line 123 "the surface area of the stream" it seems that the authors indeed mean a surface area though.

P12. Line 195: "is likely caused by pond shape, connectivity to the network and position in the network (fig 2,3,4)" I can't see from these three figures how pond shape is affecting this.

P13. In the supplementary material I see that the hydrological dominance and the biological activities are counterbalanced at a much lower hydrological dominance in large rivers than in small rivers. Could the authors explain why this is?

P14. In the supplementary material FS2 b, FS3 b, FS4 b: line is missing for large rivers?

Authors' responses to reviewers' comments to manuscript NCOMMS-17-31848-T "Thresholds of lake and reservoir connectivity in river networks control nitrogen removal"

Summary of responses by authors. The authors thank the Editors and three Reviewers for their excellent and thoughtful comments to improve the impact and clarity of this manuscript. Authors' responses are in blue text below. Reviewers' comments are labeled as the reviewer and comment number (e.g., R1-C1 is the first comment by Reviewer 1). All **bold blue text** below explicitly designates revisions to the manuscript.

We found that the reviews were extremely helpful. This revision mostly consists of clarifying the Main Text and adding supporting extensions. The clarifications to the Main Text were done to improve the presentation and better support the original conclusions. The extensions were mostly added to the Methods and Supplementary Information (SI) as supporting evidence of the original conclusions.

We have better defined and quantified thresholds where ponded waters become dominant features on the landscape throughout the Main Text: (1) a threshold in pond density (Fig. 2), (2) a threshold in the hydrologic dominance index (Fig. 3), and (3) a threshold in pond connectivity (Fig. 3) (see response to R1-C1 below). We have statistically compared the metrics reported, which now better support our original conclusions that these metrics are important to explaining where and why ponded waters are important to nitrogen removal (see response to R1-C2 and R3-C3 below). We also provided additional information on the modeling approach and limitations (see response to R2-C5 and R3-C4 below). In addition to revising the Main Text and Figures, we added a brief section to the Methods, "**Threshold estimation and statistical analysis**" (Lines 476-504). As supporting evidence, we added three tables to the SI showing the quantified thresholds (**Supplementary Tables S2 and S3**) and statistical analysis results (**Supplementary Table S4**), and a figure to more clearly show the cumulative role of ponded waters relative to streams (**Supplementary Fig. S2**; see response to R2-C1 below).

Reviewers' comments:

Reviewer #1 (Remarks to the Author):

In this paper, the authors attempt to quantify the degree to which ponds or streams dominate nitrogen removal from river networks at regional scales. The authors also use a number of landscape metrics to explain the degree to which streams or ponds dominate nitrogen removal. Nitrogen removal in both streams and ponds is estimated using the popular SPARROW model. Landscape metrics are estimated using spatial data and GIS techniques.

The authors find that the pond density, stream order, drainage area, and degree of connectivity all influence the degree to which nitrogen removal is influenced by ponds and streams. The authors also provide a novel metric called the hydrological dominance index. This index is based on theory borrowed

from chemical engineering and applied to environmental chemistry. This novel metric also explains whether nitrogen removal is dominated by streams or ponds.

Overall I found the study to be well designed and the methods appropriate. The findings are quite novel and interesting, and deserve to be seen by hydrologists everywhere. The threshold relationship between some of the metrics and pond or stream dominance of nitrogen removal is particularly interesting and novel. However, I do think that a little more is needed before this paper deserves to be in a Nature journal. I will lay out my arguments below and the editors may make the final decision.

R1-C1: The main shortcoming is that a large number of landscape factors, some novel, are applied to explain variability of pond vs stream dominance of nitrogen removal. The plots are compelling and beautiful. However, I wanted a quantitative basis for many of the statements the authors make. First, they find a number of thresholds in the relationships between variables. Yet no threshold is quantified. Quantifying a threshold is a fairly straightforward statistical operation that should be done here.

Accepted. We explicitly quantified and better define thresholds throughout the Main Text and have added a brief section to the Methods describing the details, “**Threshold estimation and statistical analysis**” (Lines 473-502). There are three thresholds in physical metrics that we define in the text that are important to determine when ponded waters are important to nitrogen removal: (1) a threshold in pond density (Fig. 2), (2) a threshold in the hydrologic dominance index (HDI; equation (2)), and (3) a threshold in pond connectivity (Fig. 3). We briefly describe below how each were quantified.

For (1) a threshold in pond density, we applied a non-parametric regression with a locally-weighted scatterplot smoothing per stream order (Fig. 2) and estimated the upper and lower limits of this regression with the positive (upper) and negative (lower) residuals. Accordingly, we define the threshold at the pond density value where the lower limit is consistently above zero. We updated Fig. 2c and 2d by adding the upper and lower limits and vertical lines explicitly showing the pond density thresholds per stream order, which are also included in a new table (**Supplementary Table S2**). We present the average pond density in the Main Text and refer the reader to Supplementary Table S2 for per stream order estimates (Lines 110-112).

For (2) a threshold in the hydrologic dominance index (HDI), we added a clarifying statement to the Main Text, “**Equation (2) shows that the threshold where a ponded water dominates nitrogen removal over its stream replacement occurs at $HDI = \frac{v_s}{v_p}$, which is where $\frac{Da_p}{Da_s} = 1$ and the proportion of nitrogen removed is equivalent between the ponded water and stream.**” (Lines 139-142).

For (3) a threshold in pond connectivity, we quantified a breakpoint in pond connectivity where there was a significant change in slope between pond connectivity and the difference in proportion removed. In addition to adding details to the “**Threshold estimation and statistical analysis**” in the Methods (Lines 486-490), we modified a statement in the Main Text, “**We found that the connectivity threshold (see Methods), where nitrogen removal by a ponded water relative to its stream replacement substantially decreases, occurs at a pond connectivity value of 0.36 (± 0.09) for NE (Fig. 3) and 0.19 (± 0.07) for CB (Supplementary Fig. S3 and Table S3).**” (Lines 157-160).

R1-C2: Second, the authors leave the reader with the sense that there are a lot of metrics that work. But which one, if any, is better? Surely a quantitative comparison is warranted here. Why not calibrate a simple statistical model to estimate how much of the variability in pond vs. stream dominance of nitrogen removal is explained by each factor? It should also be possible and straightforward to assemble multi-

variate models that explain the proportion of variance explained by combinations of factors. Indeed, the authors actually make a statement to the effect that the factors in Figure 3 explain more variance than the factors in Figure 2, yet no variances are quantified anywhere. In my mind, this lack of statistical analysis falls below the standard of due diligence of Nature.

Accepted. We calibrated a simple multilinear model to estimate how much variability in pond-stream dominance in nitrogen removal is explained by physical metrics, with details added to a new section in the Methods, “**Threshold estimation and statistical analysis**” (Lines 491-502). The important metrics we identified and currently not directly evaluated by the SPARROW model are (1) location (drainage area), (2) pond connectivity, and (3) shape (circularity), which were revealed only by comparing the existing river network to one with ponded waters replaced with streams. Through this multilinear model, we test how much variability in pond-stream dominance of nitrogen removal can be explained by these readily available physical metrics. We added a few statements to the Main Text, “**We found that pond location within the network (cumulative upstream drainage area), pond connectivity, and pond roundness explain nearly 70% of variance in the pond-stream dominance of nitrogen removal in CB and 60% in NE (see Methods and Supplementary Table S4). For both CB and NE, drainage area explains a majority of the variability because it is correlated with the HDI (Spearman rank correlation $\rho = -0.66, p < 0.001$). Rivers become larger lower in the network, generally reducing the HDI and, therefore, the dominance of a ponded water. Pond connectivity and roundness in CB explain roughly 12% and 7% of the variability, respectively. In NE, pond connectivity is slightly more important than roundness, explaining roughly 12% of the variability compared to 4% for roundness. Less connected and rounder ponded waters tend to increase the HDI. The most important physical metric for assessing pond-stream dominance is location because it is the largest control on the HDI, followed by connectivity and shape that together significantly contribute to better explaining how ponded waters function relative to stream replacements.**” (Lines 169-181). We also looked at different combinations of variables (i.e., drainage area with connectivity, drainage area with circularity), but do not report those results (percent variance explained) because they were generally consistent with the model including all three metrics.

R1-C3: I also thought that there have been a few papers recently published on similar topics that could be referenced here. They are:

Biogeochemical hotspots: Role of small water bodies in landscape nutrient processing. FY Cheng, NB Basu. *Water Resources Research*

Do geographically isolated wetlands influence landscape functions? MJ Cohen, IF Creed, L Alexander, NB Basu, AJK Calhoun, C Craft, ...*Proceedings of the National Academy of Sciences* 113 (8), 1978-1986

Geographically isolated wetlands are important biogeochemical reactors on the landscape. JM Marton, IF Creed, DB Lewis, CR Lane, NB Basu, MJ Cohen, CB Craft. *BioScience* 65 (4), 408-418

A better situation in the existing literature would better establish the need for this study.

Accepted. We have added these references to better position our study in the existing literature.

If these issues are resolved, and they should be fairly straightforward to resolve, then I believe the paper would be publishable in *Nature Communications*. Thank you for giving me the opportunity to review such an interesting paper!

Reviewer #2 (Remarks to the Author):

This is an interesting and well-written study that shows that when ponded waters replace channelized waters, nitrogen removal tends to increase, but that there is variability in the increase caused by pond characteristics. This result was obtained using the SPARROW model calibrated to the Chesapeake Bay and New England regions. The approach was to parameterize SPARROW with ponds based on existing spatial data, and then in a scenario, replace ponded waters with their underlying channels. The study also looks at various ponded water attributes, including their position in the network, degree of connectivity, and size relative to drainage area, identifying these are a source of considerable variability in N removal, beyond just the ponds hydraulic load. The major take home message is that ponded waters, and their geometry, should be considered when developing management strategies to combat nutrient pollution.

Conceptual Comments

R2-C1: The analysis, results, and presentation seem mostly sound (with some technical questions below). However, the finding that ponded waters tend to have more removal than streams they replace does not seem particularly novel. I was surprised that the network scale role of ponded waters (as opposed to local role of ponded waters replacing local stream channels) was not emphasized more. It seems like the more important points of the analysis are not really emphasized - really only mentioned briefly at one point in the main text. The results on cumulative role of ponded waters at scale of the entire region show that ponded waters as a whole, contribute very little to aquatic N removal compared to channelized streams and rivers (Table S1). Removal by the network with streams only was 47.4% and 22.2%, and with all ponds added, removal increases by <2%. In the scenario with ponds, they represent a slightly larger component of removal by the entire network (15.7% and 4.5%), but that must be because removal by ponds simply replaces removal by streams (either locally, or because downstream rivers could remove excess from upstream if ponds are removed). This is very interesting and useful result and should be emphasized and explained further.

Accepted. We agree that in addition to explaining the local controls on how ponded waters function differently than streams in nitrogen removal, an important emphasis is the relative contribution of ponded waters throughout the river network. We better emphasize and clarify the cumulative role of ponded waters in nitrogen removal throughout the Main Text, modified Supplementary Fig. S1 for a better comparison streams and ponded waters throughout the river network, and added a new figure to the SI (**Supplementary Fig. S2**) to more clearly show the cumulative role of ponded waters relative to streams for both sub-regions. Accordingly, we added a few statements to the main text, “**A lower pond density threshold suggests that it takes fewer ponded waters to cause a cumulative effect in CB. Yet the role of ponded waters in nitrogen removal decreases with increasing stream order more drastically in CB than in NE. Blending with lotic processes is more extreme in CB than in NE, obscuring the cumulative effect of ponded waters (Supplementary Fig. S2), as indicated by the cumulative stream surface area becoming greater than that of ponded waters lower in the network in CB but not in NE (Supplementary Fig. S1).**” (Lines 112-118), and “**For example, management of ponded waters in CB may have marginal improvements lower in the network due to cumulative surface areas of streams increasing faster than that of ponded waters (Supplementary Fig. S1 and S2). Although management of ponded waters in CB may not substantially reduce delivery of nitrogen to coastal waters, the metrics we identified are useful for management at local scales and within nested basins.**” (Lines 245-251). While ponded waters more frequently removed more nitrogen than a stream replacement at the local scale, as is expected, the cumulative role of ponded waters decreases with stream order due to the increased blending with lotic processes. We emphasize this point by modifying Supplementary Fig. S1 to include the cumulative surface area of streams compared to ponded waters, and developed a new cumulative hydrologic dominance index (also see related responses to R2-C2, C3, C4, and C20 below).

We have updated the values in Supplementary Table S1 after making a slight adjustment to the model calibration (changing pond removal to an exponential form (see response to R2-C29 below) and including air temperature for better consistency with previous models). None of the patterns or conclusions have been affected, but the adjustment to the model calibration provides slightly better model performance and slightly more removal by ponded waters. We now compare the model calibration estimates to literature values to better place modeling results in the appropriate context by adding **Supplementary Fig. S6** (also see response to R2-C2, C7, and C15 below).

R2-C2: If it is the true that rivers dominate network scale removal, then why is that the case? Their biological activity as uptake velocity (Table S2) is about 10x higher than in ponds (at least in low orders). This is also an interesting result that is not really emphasized at all. It would be helpful to also report the total surface area of streams and rivers in each region (with and without ponds).

Accepted. Rivers tend to dominate network-scale removal due to increased lentic and lotic blending; however, portions of the network are clearly dominated by ponded waters and should not be overlooked. We modified Supplementary Fig. S1 to include the cumulative surface area of streams and better explain why streams and rivers, for the most part, dominate network-scale removal. The lotic and lentic blending is also quantified with the new cumulative hydrologic dominance index (HDI) (equation (6); see response to R2-C20 below). We better emphasize cumulative removal throughout the Main Text, for example, **“The cumulative HDI (see Methods) is a good indicator of how the entire river network is affected by replacing ponded waters with streams (Fig. 1a, 4a) because this metric, unlike pond density, better reflects the balance of lotic and lentic blending.”** (Lines 202-204).

We also better emphasize the importance of biological activities in the Main Text, **“As shown by a Damköhler analysis, the underlying reason is that the threshold in the HDI where a ponded water becomes dominant will shift directly as a function of the ratio of biological activities. Streams are up to 18 times more efficient than ponded waters in NE and 14 times more efficient in CB (Supplementary Table S5), resulting in a slightly larger pond-dominance threshold in HDI for NE. While the calibrated biological activity estimates are considered reasonable (i.e., within the range of empirical values from the literature; Supplementary Fig. S6), possible variation in the ratio of biological activities is likely to be much less than the range of HDI throughout the river network. For example, ponded waters along headwaters have surface areas (and corresponding residence times) much larger than their stream replacements that overwhelm the difference in biological activities, in contrast to ponded waters lower in the network (Supplementary Fig. S5).”** (Lines 186-196).

R2-C3: The fact that stream reactivity appears to be 10x high in streams and rivers compared to ponds (which is counter to some examples reported in the literature) and that as a result the cumulative ponded role at network scale is small, seems to me to be more relevant regional result that SPARROW is ideally suited to demonstrate.

Accepted. We agree that the results from SPARROW indicate that the cumulative role of ponded waters is marginal in CB and notable in NE and have added and revised text to emphasize that important point (see responses to R2-C1 and R2-C2 above, and R2-C4 below). However, one of the main goals of this paper is to explain where and why ponded waters are important, which we emphasize by finding location, connectivity, and shape are important. We have clarified that while ponded waters may not be dominant to network scale removal, particularly in CB, the local analysis is still useful, for example, **“Although management of ponded waters in CB may not substantially**

reduce delivery of nitrogen to coastal waters, the metrics we identified are useful for management at local scales and within nested basins.” (Lines 249-251).

R2-C4: The current emphasis on what happens if ponded areas replace streams is still useful, but if ponds as a whole are not a big factor, will managing them (e.g. via reservoirs as discussed later in the text) really make a difference? Any one reservoir seems it would have little impact, in larger rivers because dominance index is small and not affected by shape, and in headwaters because you would need very many of them as any one will intercept little N load. This has implications for which kinds of dam removals to emphasize, if you are factoring in their role in N removal in whether to take down.

Accepted. We agree that it is important to point out to the reader that if ponded waters have little influence on the cumulative removal, managing them may have a marginal impact. We modified a few statements in the Main Text to better emphasize this important point, **“Striking regional differences confirms a need to determine how and where the management of ponded waters can reduce downstream nitrogen loading.”** (Lines 99-101). We also revised a discussion statement, **“Our results indicate that small ponded waters on headwater streams, farm ponds or beaver ponds for example, can sometimes be effective as restoration tools^{16,32,33} by reducing downstream nitrogen loading, but only if the blending with lotic process does not overwhelm and obscure the effect.”** (Lines 253-256). Also see related responses to R2-C1, C2, and C3 above.

Technical Comments

R2-C5: I could not figure out how pond characteristics other than uptake velocity and pond surface area could be assessed in the SPARROW model as described in the text. The importance of pond characteristics is a major point of the paper. Pond connectivity in particular was identified as an important secondary factor after hydraulic load. The metric for connectivity indicates that ponds that are little connected have less removal (as in bottom right inset in figure 3). But if that pond in Figure 3 is replaced by a stream, then the stream is very short ($L \cdot b$ is small compared A of the pond). The SPARROW model assumes all ponded surface areas have the same reactivity (L354) and all ponds assumed to be completely mixed (L372). The model is calibrated with only v_f and HL in the removal equations. So how does connectivity factor in? I may be missing something very simple, but I think this should be addressed clearly and explicitly in the methods to increase understandability for most readers.

Accepted. Pond connectivity was not brought directly into the SPARROW model. The question **“Where and why are ponded waters important?”** was addressed through the comparison of the existing network to one with ponded waters replaced with streams. Only by looking at removal by a ponded water relative to a stream replacement did we evaluate that location, connectivity, and shape are important—a disconnected ponded water (not centered) has less of an influence on nitrogen removal relative to a stream. You are correct that all ponded waters are assumed equally reactive within the SPARROW model, but through our comparison it is clear that connectivity and shape are important considerations to the management of ponded waters. Following a statistical analysis (see response to R1-C2 above), we added clarifying text, **“Less connected and rounder ponded waters tend to increase the HDI. The most important physical metric for assessing pond-stream dominance is location because it is the largest control on the HDI, followed by connectivity and shape that together significantly contribute to better explaining how ponded waters function relative to stream replacements.”** (Lines 177-181). Furthermore, to be cleared to the reader, we added a statement to the Main Text, **“However, the underlying causes of reduced cumulative removal with increasing stream order for a similar pond density and differences between sub-regions were only recognized by analyzing the balance of hydrologic and biological factors at the local scale.”** (Lines 220-223) and to the Methods, **“Metrics of pond shape and connectivity were not brought directly into the SPARROW model; they were found important only by comparing the two cases.”** (Lines 309-311).

R2-C6: I was also confused by some of the terminology. Does the term "cumulative nitrogen removal" (L96, L184) refer to total nitrogen removal within an NHD subcatchment (I think it does based on the figures 1b and 2), but I would normally interpret cumulative as meaning total removal by all lakes in the upstream catchment. This term should be clearly defined in this context. Is this term the same as "cumulative effects" in larger river basins (L101)? Is this meant to be cumulative effects in all large rivers (which is sum of local) or entire river basin (which is sum of all upstream). I would review the use of these terms to make sure they are clear.

Accepted. We make a clearer distinction between “local” and “cumulative” throughout the Main Text. Indeed, “cumulative nitrogen removal” refers the total removal by all ponded waters and streams upstream of that catchment, and the cumulative results are presented in Fig. 1 and 2 (labels has been updated to “cumulative”). We have carefully clarified this term throughout the Main Text, “Our results showed that the effect of ponded waters on cumulative nitrogen removal—**estimated from headwaters to coasts as the total amount removed upstream of each location**—varies throughout the river network (Fig. 1a; see Methods).” (Lines 89-91), which is separate from the local (or individual) removal within a single catchment.

R2-C7: Another was the term used for the aquatic decay parameters in Table S2. I think these should be reporting uptake velocity (as discussed in the main text), but in S2 the "inverse hydraulic load" is reported. L373 points out the reciprocal hydraulic load, but this is not what is in Table S2. Hydraulic load is not the decay coefficient and I don't think is calibrated, but is instead specified based on channel characteristics (discharge, width, length). I think this is really biological reaction time scale? But I would report as uptake velocity, as it is more intuitive and reported in field studies. Comparison with empirical measurements of this result, as done one of the early Alexander et al. 2000 papers would be helpful to place these modeling results in appropriate context.

Accepted. It is common practice to present SPARROW calibration results by the explanatory variable. For calibration of the “uptake velocity” as stated in the text, the explanatory variable is the reciprocal hydraulic load. For consistency with the literature, we have keep the explanatory variables as is in Supplementary Table S5 (was Table S2), but modified the columns of Table S5 to clarify “coefficient units” associated with the uptake velocity. We also added a statement to the bottom of the table indicating that the coefficient estimates represent nitrogen uptake velocities. In addition, we now compare the model calibration estimates to literature values to better place modeling results in the appropriate context by adding **Supplementary Fig. S6** (also see response to R2-C2 above).

Specific Comments

R2-C8: L56. Say depth and residence time. (Volume to surface area ratio = depth)

Accepted. We modified the text, “...residence and **depth** (the volume to benthic surface area ratio).” (Lines 56-58). We feel that it is important to remind the reader that depth comes from the surface area to volume ratio.

R2-C9: L84-85. Is creating more dams likely? It would probably be worth mentioning the trend of dam removals due to age occurring in the region, and debates on whether to replace.

Accepted. We added a statement and relevant literature to the discussion rather than here, “**While the number of large reservoirs is likely to increase worldwide²⁹, the more likely trend in the United States is dam removal due to age³⁰.**” (Lines 240-242).

R2-C10: L105-106. Can you be more explicit on how it explains?

Accepted. We modified and clarified this statement for a better transition to the next paragraphs, “**We focused on the individual (local) balance of hydrologic and biological factors to better explain why headwater ponded waters have the largest influence, and why there are differences in pond density thresholds across stream order and between the two sub-regions.**” (Lines 119-121).

R2-C11: L122. I think that it would be helpful to note that the hydrologic ratio will always be greater than one, but that the size of the ratio determines the hydrologic dominance.

Acknowledged. We feel that such a statement here will introduce confusion for the reader. We think it is better to emphasize the story of “hydrologic and biological tradeoffs” and accomplish the meaning by, “As the HDI increases, the dominance of the ponded water over its stream replacement increases (Fig. 3).” (Lines 142-143).

R2-C12: L147. In text, threshold is 1, but in figure, it is 0 (log of 1). I would make consistent.

Accepted. We updated the label in the Fig. 3 scale from log transformed to absolute.

R2-C13: L157. What is meant by cumulative effects? Within NHD subbasin or in entire upstream watershed, or in entire region? I suggest being more specific.

Accepted. Please also see response to R2-C6 above. Cumulative effects at a point in the network are the sum of everything upstream. Throughout the Main Text and Methods we make a clearer distinction between **cumulative as the sum of everything upstream** and **individual (local)** that corresponds to a single NHD catchment.

R2-C14: L161. Tie this back to total removal by lakes in entire river network. Even though NE has fewer ponds that are dominant, because there are more of them (or more total surface area), ponds play a more important role in NE than in CB.

Accepted. Please also see response to R2-C1 above. We revised and added statements throughout the Main Text to better emphasize removal from the entire river network, for example, “**A lower pond density threshold suggests that it takes fewer ponded waters to cause a cumulative effect in CB. Yet the role of ponded waters in nitrogen removal decreases with increasing stream order more drastically in CB than in NE. Blending with lotic processes is more extreme in CB than in NE, obscuring the cumulative effect of ponded waters (Supplementary Fig. S2), as indicated by the cumulative stream surface area becoming greater than that of ponded waters lower in the network in CB but not in NE (Supplementary Fig. S1).**” (Lines 112-118), and, “**This balance of hydrologic and biological factors affecting nitrogen removal at the local scale can be extended to provide a simple cumulative metric that allows for quick assessment of where and why ponded waters affect nitrogen removal throughout the entire river network, which will be important to evaluating the potential effectiveness of ponded water management on downstream nitrogen loading.**” (Lines 197-201).

R2-C15: L168. This difference in biological activity between streams and ponds should be emphasized much more, including relating to empirical measurements in the literature. Is this consistent with empirical observations? Rivers and ponds are less different (if you average NE and CB). The difference between ponds and small streams is much greater than difference between ponds in CB and NE (only 2x different).

Accepted. Please also see responses to R2-C2 and R2-C7 above. We now relate calibrated values to empirical measurements from the literature. We added a figure to the SI (**Supplementary Fig. S6**) showing the nitrogen removal rates by streams and ponded waters as a function of water depth. We added a statement to the Main Text, **“Streams are up to 18 times more efficient than ponded waters in NE and 14 times more efficient in CB (Supplementary Table S5), resulting in a slightly larger pond-dominance threshold in HDI for NE. While the calibrated biological activity estimates are considered reasonable (i.e., within the range of empirical values from the literature; Supplementary Fig. S6), possible variation in the ratio of biological activities is likely to be much less than the range of HDI throughout the river network.”** (Lines 188-194).

R2-C16: L184. This should clearly state that ponds dominate over their stream replacement only, not at network scale (statement could be misinterpreted as written).

Accepted. Please also see response to R2-C1 above. We agree and the text has been revised to prevent misinterpretation of the cumulative role of ponded waters, **“In CB, pond density decreases with stream order, resulting in marginal coastal delivery reduction from ponded waters (Supplementary Figs. S1 and S2). Conversely, pond density remains relatively constant in NE, corresponding with notable cumulative removal and reduced coastal delivery from ponded waters (Supplementary Figs. S1 and S2). However, the underlying causes of reduced cumulative removal with increasing stream order for a similar pond density and differences between sub-regions were only recognized by analyzing the balance of hydrologic and biological factors at the local scale.”** (Lines 216-223).

R2-C17: L197. But if ponded waters have such a small role at the scale of entire network, will management of individual ponded water bodies really be effective?

Accepted. Please also see responses to R2-C1, C2, C3, and C4. We added a statement important to the reader here to point out that regional management may be marginal low in the network, but may still be effective at local and nested basin scales, **“...management of ponded waters in CB may have marginal improvements lower in the network due to cumulative surface areas of streams increasing faster than that of ponded waters (Supplementary Fig. S1 and S2).”** (Lines 245-248).

R2-C18: L205. Improve water quality for N, but maybe not for other things (use of DO, pathogens, etc).

Accepted. We changed “water quality” to **“nitrogen loading”** (Line 255).

R2-C19: L261. Is pond density for the entire upstream drainage area (which I would call cumulative) or just for the NHD subcatchment. Its described slightly differently in different places. L265 suggests it is for subcatchment. Figure S1d suggests it is per stream order. Please clarify.

Accepted. It is indeed the entire upstream drainage area (and the sum of all upstream ponded water surface areas). This has been clarified, **“...within some of the upstream NHD catchments that contribute to the summation of cumulative drainage area.”** (Lines 338-339).

R2-C20: L284. This index is only about those stream reaches that have ponds, and not all stream reaches, right? Because it is stated that the index is applied to the entire network, it could be interpreted as the relative importance of ponds vs. all streams. Based on results in Table S2, ponds are relatively unimportant. I would clarify in this section that this is the case. Also, L171 is a bit misleading because it says that this index is a good indicator of pond dominant parts of the network. This has not been demonstrated because it only talks about those reaches that have ponds and not all the connecting reaches that have no ponds.

Accepted. Yes you are correct, this metric is estimated only for stream reaches that have ponds. We agree that a metric that includes all stream reaches would be better. Therefore, we developed the cumulative hydrologic dominance index (equation (6) in the Methods), which now reflects the complete blending of lotic and lentic waters. This metric was updated to Figure 4a, which provides an indication to which parts of the river network are affected by ponded waters and illustrates why management of CB pond waters may not have notable effects on loads delivered to the coast. We revised the Main Text here, **“The cumulative HDI (see Methods) is a good indicator of how the entire river network is affected by replacing ponded waters with streams (Fig. 1a, 4a) because this metric, unlike pond density, better reflects the balance of lotic and lentic blending.”** (Lines 202-204) and **“Furthermore, less removal for a given pond density is due to a lower cumulative HDI, reflecting a greater degree of lotic blending and reduced effect of ponded waters.”** (Lines 213-214). We also updated the details to developing this metric in the Methods. For example, **“By comparing the existing and pond-replaced river networks, the cumulative HDI indicates portions of the network that are either hydrologically dominated by ponded waters or streams...Portions of the network where ponded waters dominate nitrogen removal are, therefore, generally indicated by a cumulative HDI that is greater than the ratio of biological activities $\left(\frac{v_s}{v_p}\right)$.”** (Lines 357-365).

R2-C21: L293. Explain purpose of this analysis in opening sentence (i.e. estimate connectivity).

Accepted. We added a topic sentence to explain the purpose, **“Pond connectivity was estimated to assess the degree of centeredness of a ponded water on the river network.”** (Lines 366-367).

R2-C22: L306. Is perimeter of the ponded water in the NHD plus? I don't think this attribute was stated when discussing the data set.

Accepted. Perimeter is not an attribute provided within NHD. We added a statement to clarify that we estimated this, **“Perimeter was estimated directly from the polygons provided by NHD.”** (Lines 326-327).

R2-C23: L390-399. Confusing. How is equation sensitive to circularity or connectivity?

Accepted. We agree that showing this equation here adds confusion and have deleted it because it was not used in any of the calculations or results. In its place, we added a clarifying statement to the Main Text, **“Less connected and rounder ponded waters tend to increase the HDI. The most important physical metric for assessing pond-stream dominance is location because it is the largest control on the HDI, followed by connectivity and shape that together significantly contribute to better explaining how ponded waters function relative to stream replacements.”** (Lines 177-181).

R2-C24: L407-408. Variable abbreviation identical.

Accepted. Nice catch. This has been fixed to specific accumulated load out of ponds and streams (Lines 464-465).

R2-C25: Figures. The vertical lines in Figure 3 and S 2-4 show the v ratio where removal by ponds is greater than streams. It took me a while to understand what the vertical lines represent and should be explained more clearly. I think an arrow pointing at the X axis value of hydrologic ratio could be better.

These figures captions should define that the ratio of biological activity has same units hydrologic index, and is the value of hydro index when role of ponds and streams are equal.

Accepted. Yes, the vertical line (changed to a vertical arrow in Fig. 3) comes from when removal by ponded waters and streams are equal, which is now better reflected in the Main Text, “**Equation (2) shows that the threshold where a ponded water dominates nitrogen removal over its stream replacement occurs at $HDI = \frac{v_s}{v_p}$, which is where $\frac{Da_p}{Da_s} = 1$ and the proportion of nitrogen removed is equivalent between the ponded water and stream.**” (Lines 139-142). We also better state this in the caption of Fig. 3, “**The threshold in hydrologic dominance index where ponded waters becomes dominant occurs when it is equal to the ratio of stream to ponded water biological activities, which results from the case when the associated Damköhler numbers are equal (see equation (2)).**”

R2-C26: Figure 1B - show scale values not in log.

Accepted. The scale in Fig. 1b has been changed from log to absolute values.

R2-C27: Figure 3. Mismatch in use of pond density in terms of % in text and log proportion in figure. Make consistent.

Accepted. We have updated pond density as a % in the text and in Fig. 2.

R2-C28: EQN 6 - is this only of reaches with ponds? If so, its not really an index of the whole watershed relative importance of ponds to stream reaches. I would rather see an index of the whole network, not just the reaches with ponds.

Accepted. Please see response to comment R2-C20 for details. Yes, this previous metric is based on only the summation of the streams reaches with ponds. We agree that a cumulative metric that includes every stream reach is more effective. Therefore, we have updated this metric as the cumulative hydrologic dominance index (equation (6) in the Methods), which is the summation of all wetted surface areas in the existing river network (both existing ponds and streams) relative to the summation of all wetted surface area of the network where ponded waters are replaced with streams (all existing streams and stream replacements). We agree that this provides a better representation of how a ponded water will effect removal when there is accumulated blending with streams in large river basins.

R2-C29: EQN 9 and 10. Why use different equations? Why not use exponential form for ponded? Seems more parsimonious to use the same equation.

Accepted. We agree and have updated the equation so we use a consistent exponential forms for streams and ponded waters.

R2-C30: Supplemental Figures 2-4. Alter titles. All main titles are identical and it takes a while to figure out what is different about each one.

Accepted. We have altered the titles to be more explicit to what each one represents.

Reviewer #3 (Remarks to the Author):

The manuscript entitled “Thresholds of lake and reservoir connectivity in river networks control nitrogen removal” focusses on effect of different characteristics of ponded waters on nitrogen removal. What is particularly new, is that ponded waters are compared with their stream replacements. Such comparison is of great value to managers that want to construct or remove reservoirs e.g. for clean drinking water provisioning. The main conclusion is that pond connectivity and hydrologic dominance index have most effect on nitrogen retention.

The comparison between hydrological and biological factors is intriguing. At the same time the current introduction of these factors makes things complicated to understand (see my point P1). Furthermore I had some other comments and suggestions on the definitions (P2) and conclusions (P3). After my main points I have some other suggestions and comments.

Main points:

R3-C1. P1. The hydrologic dominance index is a nice factor to explain the effect of hydrology. However, it took me a while to understand the vertical line representing the biological factors as described by $v_s:v_p$ in most of the figures. After rewriting several equations I found the logic behind it (at least I think so). The understanding of the $v_s:v_p$ is important to understand the thresholds that are described in the paper. I would therefore suggest that the authors clarify the manuscript and figures with respect to the thresholds and the equations, preferably already in the main text (before the methods).

Accepted. We have revised the Main Text to better clarify, “**Equation (2) shows that the threshold where a ponded water dominates nitrogen removal over its stream replacement occurs at $HDI = \frac{v_s}{v_p}$, which is where $\frac{Da_p}{Da_s} = 1$ and the proportion of nitrogen removed is equivalent between the ponded water and stream.**” (Lines 139-142). We also moved the Damköhler numbers to the Main Text (equation (1)) to more clearly show where the hydrologic dominance index comes from. Lastly, we clarified the Fig. 3 caption, “**The threshold in hydrologic dominance index where ponded waters becomes dominant occurs when it is equal to the ratio of stream to ponded water biological activities, which results from the case when the associated Damköhler numbers are equal (see equation (2)).**”

R3-C2. P2. I had difficulties with understanding the term ‘threshold’. After reading the title it was not yet clear to me what kind of thresholds the authors meant. As connectivity was mentioned in the title I was thinking in the direction of hydrological connectivity thresholds. Then two new terms in the abstract (“thresholds in pond density” and “cumulative thresholds”) did confuse me as I missed the term “connectivity” in relation to “threshold”. After reading the manuscript I had the feeling there were many thresholds, in size, shape, connectivity, position and density, some of them being more important than others. I would suggest that the authors would be more clear in the title, abstract and introduction about what kind of threshold(s) they are looking at.

Accepted. Throughout the Main Text we better clarify and define thresholds: (1) threshold in pond density as a cumulative metric, (2) threshold in the hydrologic dominance index when the removal of an individual pond and a stream replacement are equal as indicated by equation (2) (see response to R3-C1 above), and (3) threshold in pond connectivity where the effect of an individual pond relative to its stream replacement starts to decrease. We revised the Abstract to clarify the thresholds we quantify, “Thresholds in pond density **where ponded water become important to regional water quality outcomes...**” and “...along with thresholds **in connectivity to the network...**” (Lines 26-29 and 31).

R3-C3. P3. The conclusions of the authors are that the different factors (e.g. size, shape, connectivity, position and density) are not equally important in the effect on nutrient retention. (see for example line 153 onwards: “While the shape of a ponded water is clearly important... water versus streams”. How is it tested that the shape of a ponded water has less effect on nitrogen removal than pond connectivity and hydrologic dominance index? I missed the statistical explanation on which this statement is based.

Accepted. We quantified clear thresholds in pond density that indicate when ponded waters influence regional water quality outcomes (Fig. 2), and then shift our focus to, “...**the individual (local) balance of hydrologic and biological factors to better explain why headwater ponded waters have the largest influence, and why there are differences in pond density thresholds across stream order and between the two sub-regions.**” (Lines 121-121). For a statistical evaluation of the variance explained by physical metrics and relative importance, we applied a multilinear model, with details added to a new Methods section, “**Threshold estimation and statistical analysis**” (Lines 476-504). The important metrics we identified that are not directly evaluated by the SPARROW model are (1) location (drainage area), (2) pond connectivity, and (3) shape (circularity), which were revealed only by comparing the existing river network to one with ponded waters replaced with streams. We added a summary of results to the Main Text, “**We found that pond location within the network (cumulative upstream drainage area), pond connectivity, and pond roundness explain nearly 70% of variance in the pond-stream dominance of nitrogen removal in CB and 60% in NE (see Methods and Supplementary Table S4). For both CB and NE, drainage area explains a majority of the variability because it is correlated with the HDI (Spearman rank correlation $\rho = -0.66, p < 0.001$). Rivers become larger lower in the network, generally reducing the HDI and, therefore, the dominance of a ponded water. Pond connectivity and roundness in CB explain roughly 12% and 7% of the variability, respectively. In NE, pond connectivity is slightly more important than roundness, explaining roughly 12% of the variability compared to 4% for roundness. Less connected and rounder ponded waters tend to increase the HDI. The most important physical metric for assessing pond-stream dominance is location because it is the largest control on the HDI, followed by connectivity and shape that together significantly contribute to better explaining how ponded waters function relative to stream replacements.**” (Lines 169-181). Also see response to R1-C2 above.

Other points:

R3-C4. P4. What are the main pitfalls of the used modelling approach including the stream replacements? How do they influence the results?

Accepted. The main pitfall with modeling with the stream replacements is an accurate estimate of channel length, which affects the stream travel time. We use the reach length estimates provided by NHD, which are estimates of the most direct path a channel might have through the ponded water. We added a few statements to the Methods acknowledging potential pitfalls, “**These flowlines may be artificially short due to lower than natural sinuosity (approaching one), which would reduce the estimated stream travel time and, therefore, lessen nitrogen removal by the stream replacement. As a test, increasing the sinuosity of these flowlines to two for a typical meander results in a stream travel time increase by a maximum factor of two, assuming velocity remains unchanged. Because ponded waters generally have reciprocal hydraulic loads much larger than their stream replacements (on average 7.5 times larger), doubling the stream travel times does not change the overall patterns of results or conclusions. Furthermore, we anticipate that minor adjustments to sinuosity have negligible influence on pond connectivity estimates because artificial flowlines represent the most direct downstream path.**” (Lines 408-417).

As shown by the figure below, doubling reach travel times (which we consider an extreme case) causes the dominance of a stream relative to a ponded water to increase, but the overall pattern and

conclusions remain the same. There is a slight shift in the hydrologic dominance index, but because ponds are typically much larger than stream replacements, the original pattern remains. Therefore, we conclude that the revised statement above is accurate and sufficient.

R3-C5. P5. Line 36: what is meant by accumulated biogeochemical processes?

Accepted. We refer to “biogeochemical processes” as the suite of processes that act on the transformation of nutrients and water quality, which is common terminology in the literature. We elect to keep “biogeochemical processes” to acknowledge that there are a host of processes acting on water quality, but have revised the text for clarity, “...which store, convey, and transform mass and energy through accumulated biogeochemical **processing of transported materials...**” (Lines 35-37).

R3-C6. P6. Line 42 and onwards: According to the authors high nutrient retention in constructed reservoirs is an issue because of the potential of cyanobacterial blooms and related toxins, while this seems not been recognized as an issue in a constructed or natural pond. It is not clear to me how this differences come about.

Accepted. We have clarified this difference in the text, “..., **may have a more favorable balance of hydrologic and biological functions** that benefit water quality.” (Lines 49-50). In the following paragraph we specify that the balance of conditions is controlled by the residence time and the surface area to volume ratio.

R3-C7. P7. Line 48: what is meant by biogeochemical hotspots?

Accepted. By addressing the previous comment and clarifying why constructed ponds can benefit water quality (see response to R3-C6 above), we have deleted “biogeochemical hotspots” here because it is no longer necessary.

R3-C8. P8. Line 53: ‘Although’ suggest a contradiction, however I don’t understand the contradiction. The first part of the sentence seems to contain the same information as the second part.

Accepted. We have clarified the contradiction, “**Although the relative roles of streams and ponded waters may vary throughout river networks and across regions**, their individual rates of nitrogen removal often vary consistently.” (Lines 54-56).

R3-C9. P9. Line 104 onwards: “In addition to ... two regions.” I don’t understand this.

Accepted. We modified this statement for a clearer transition, “**We focused on the individual (local) balance of hydrologic and biological factors to better explain why headwater ponded waters have the largest influence, and why there are differences in pond density thresholds across stream order and between the two sub-regions.**” (Lines 119-121).

R3-C10. P10. Equation 1: if τ (tau) is the mean annual stream travel time I would think that $A*d/Q$ is the mean annual pond travel time (or better ‘residence time’). However, d seems to be the depth of the stream, and not the average depth of the lake. Could the authors clarify this more?

Accepted. Yes, $A*d/Q$ is the mean annual pond residence time. We do not include residence time of the ponded water in equation (1) (now equation (2)) like we do with the stream because there is likely more uncertainty associated with a pond depth than with a stream depth. We estimate stream depth through hydraulic geometry. We chose to use this formulation for estimating nitrogen removal because it is based on the best available data. However, in the Main Text we now use a consistent formulation for the reciprocal hydraulic load as the surface area divided by flow, which represents, “...**the standard hydraulic metric of time required to displace a unit volume of water.**” (Lines 125-126). We also added equations (1a) and (1b) for clarity (Lines 128-129).

R3-C11. P11. Additionally at equation 1, the unit of L_b ($=d[m]/(Q[m^3/d]*\tau[d])$) is meters according to the explanation, however a simple unit analysis shows that this should be m^2 , but then it could not be the length of the stream as stated in line 118. Reading line 123 “the surface area of the stream” it seems that the authors indeed mean a surface area though.

Accepted. L_b is indeed units of m^2 , which is the surface area of the stream. We have thoroughly checked all units in the explanation of equation (1) for correctness. Our formulation of equation (1) is correct (now equation (2)), and have modified it for consistency and clarity (see response to R3-C10).

R3-C12. P12. Line 195: “is likely caused by pond shape, connectivity to the network and position in the network (fig 2,3,4)” I can’t see from these three figures how pond shape is affecting this.

Accepted. Pond shape is in the Supplement. We now reference Supplementary Fig. S4 here.

R3-C13. P13. In the supplementary material I see that the hydrological dominance and the biological activities are counterbalanced at a much lower hydrological dominance in large rivers than in small rivers. Could the authors explain why this is?

Accepted. In the model calibration, we estimated separate uptake velocities for small streams and large rivers. We make this specification based on previous work in the Northeast that suggests large rivers essentially function like pipes and removed very little nitrogen. We feel that calibrating the model to small streams and large rivers is appropriate to get the best possible estimates of biological activity. We added a brief statement to the Methods, “**Small streams are statistically significant to the predicted nitrogen removal while large rivers are not (Supplementary Table S5). Therefore, to achieve the most likely estimates for nitrogen uptake in streams, we make this specification for two stream size classes.**” (Lines 532-535).

R3-C14. P14. In the supplementary material FS2 b, FS3 b, FS4 b: line is missing for large rivers?

Accepted. We have added the estimates for large rivers to these SI figures.

REVIEWERS' COMMENTS:

Reviewer #1 (Remarks to the Author):

Thank you for addressing all my comments satisfactorily. I believe the paper can now be published.

Reviewer #2 (Remarks to the Author):

The authors have thoughtfully addressed my comments. I think the paper is technically very sound, comprehensive, interesting, and well written. For the management implications, I liked the suggestion that dam removals in the US should consider the size/shape/ location/connectivity of these when considering which may be considered for removal if nutrient management is an issue. Another suggestion was to add headwater ponds in CB, but I wondered whether the authors could provide an estimate of how many would be needed to, say, match the contribution of ponds in New England (~20% of total removal there). That seems like it would be a useful management target. I like the idea of blending, but it may be helpful to have a more concrete definition of what this is.

L186-196. This makes sense, but it must only occur in a few limited locations since NE ponds contribute much more than CB ponds. What about the fact that the uptake velocities for ponds are not significantly different from 0 (based on p values)?

L189. Add "biologically" before efficient.

L219. Add parenthetical after notable: "(~20% of total aquatic removal)".

Table S5. I still find the units confusing in Table S5. Use of reciprocal hydraulic load in the variable column, but units column has units of length/time. If these are the decay rates, why is it called hydraulic load? I would still use the same units for lakes and streams, so easy to compare. Given the uncertainties in the uptake velocities (only small streams are significantly different from 0), it could be that lakes may have no net annual removal at all.

Figure S6. Given that uptake velocity is reported earlier in the paper, it would be much more useful to also have this figure with these units.

Reviewer #3 (Remarks to the Author):

The revised version of the manuscript entitled "Thresholds of lake and reservoir connectivity in river networks control nitrogen removal" has much improved. Most of comments are dealt with, except for a few minor things as described below.

R3-C1: I think the authors did a good job in clarifying this equations. I have a few minor suggestions left, though.

1) I would suggest to put a multiplication dot between L and b in all equations this is found to clarify that the b is not a subscript

2) Equation 2 can only be possible if Q_p equals Q_s . This is already stated as ("assumed equal over the annual time period"), but when I read that the first time, I read it as no variation in time within a year which is confusing. Maybe change the sentence to: "assumed equal over the annual time period, so that $Q_p=Q_s$ ".

Authors' responses to reviewers' comments to manuscript NCOMMS-17-31848A "Thresholds of lake and reservoir connectivity in river networks control nitrogen removal"

Summary of responses by authors. The authors thank the Editors and three Reviewers for again providing thoughtful and helpful comments. Authors' responses are in blue text below. Reviewers' comments are labeled as the reviewer and comment number (e.g., R1-C1 is the first comment by Reviewer 1). All **bold blue text** below explicitly designates revisions to the manuscript.

The revisions were all minor, but have further improved the clarity and impact of this manuscript. In addition to minor edits, we added a few clarifying statements. In particular, we better clarify how our newly developed physical metrics can be used to quickly assess management of ponded waters (see response to R2-C1 below).

Reviewers' comments:

Reviewer #1 (Remarks to the Author):

R1-C1: Thank you for addressing all my comments satisfactorily. I believe the paper can now be published.

No author response necessary. Thank you.

Reviewer #2 (Remarks to the Author):

The authors have thoughtfully addressed my comments. I think the paper is technically very sound, comprehensive, interesting, and well written. For the management implications, I liked the suggestion that dam removals in the US should consider the size/shape/ location/connectivity of these when considering which may be considered for removal if nutrient management is an issue.

R2-C1: Another suggestion was to add headwater ponds in CB, but I wondered whether the authors could provide an estimate of how many would be needed to, say, match the contribution of ponds in New England (~20% of total removal there). That seems like it would be a useful management target.

Accepted. To provide a useful suggestion for managing ponded waters, we added a brief statement, **"The cumulative HDI is also potentially useful to quickly assess how management of ponded waters could reduce nitrogen loading to coastal areas. For the Northeastern United States, for example, a 1% increase in cumulative proportion nitrogen removed requires roughly a factor of two increase in the cumulative HDI. A possible management strategy in CB could be to increase the cumulative HDI with small ponded waters on headwater streams, beaver ponds for example, as these may be effective as restoration tools^{16,32,33} by reducing downstream nitrogen loading."** (Lines 262-268). While we agree with your sentiment of estimating an actual number of ponded waters in CB to match removal in NE, we feel that providing an estimate of percent increase in removal per increase in HDI is much more useful and appropriate given the complex nature of nutrient management in CB.

R2-C2: I like the idea of blending, but it may be helpful to have a more concrete definition of what this is.

Accepted. We use “blending” to describe the downstream accumulation of ponded waters and streams. Therefore, we clarified the meaning, “...the blending **or downstream accumulation** of lotic and lentic processes...” (Line 115).

R2-C3: L186-196. This makes sense, but it must only occur in a few limited locations since NE ponds contribute much more than CB ponds. What about the fact that the uptake velocities for ponds are not significantly different from 0 (based on p values)?

Acknowledged. For CB, an uptake velocity of 3.6 m/y is generally consistent with previous water quality modeling studies in this sub-region (e.g., ref 43), which has a p-value of 0.056 that we consider meaningful because it is so close to the common alpha level of 0.05. Although the NE p-value is not below the alpha level of 0.05, we consider it meaningful and feel confident moving forward with the coefficient estimate because it is generally consistency with prior work in the Northeast (ref. 14).

R2-C4: L189. Add “biologically” before efficient.

Accepted. This now reads, “...more **biologically** efficient...” (Line 201).

R2-C5: L219. Add parenthetical after notable: “(~20% of total aquatic removal”).

Accepted. We revised this to, “...notable cumulative removal and reduced coastal delivery from ponded waters (**~20% of total aquatic removal**; Supplementary Fig. 1, 2).” (Line 230).

R2-C6: Table S5. I still find the units confusing in Table S5. Use of reciprocal hydraulic load in the variable column, but units column has units of length/time. If these are the decay rates, why is it called hydraulic load? I would still use the same units for lakes and streams, so easy to compare. Given the uncertainties in the uptake velocities (only small streams are significantly different from 0), it could be that lakes may have no net annual removal at all.

Accepted. Our previous explanation stands, but we see the difficulty in understanding the table. Therefore, we have updated the table for clarity by removing “reciprocal hydraulic load” and only presenting the units for the uptake velocity (length/time), which is consistent with ref. 23. We have also updated the units for stream and ponded water decay to m/y for consistency. There is larger uncertainty associated with the uptake velocity for NE ponded waters, but we consider the estimate meaningful based on previous studies in the region and literature rates (see response to R2-C3 above).

R2-C7: Figure S6. Given that uptake velocity is reported earlier in the paper, it would be much more useful to also have this figure with these units.

Accepted. Because the uptake velocity is independent of the size of a water body, we think it is useful to show how the loss rate varies with size in comparison to literature values. However, we agree that this could be source of confusion for the reader and have clarified the definition in the Supplementary Fig. 6 caption, “**Nitrogen loss rates are quantified as the uptake velocities divided by water depth.**”

Reviewer #3 (Remarks to the Author):

The revised version of the manuscript entitled “Thresholds of lake and reservoir connectivity in river networks control nitrogen removal” has much improved. Most of comments are dealt with, except for a few minor things as described below.

I think the authors did a good job in clarifying this equations. I have a few minor suggestions left, though.

R3-C1: 1) I would suggest to put a multiplication dot between L and b in all equations this is found to clarify that the b is not a subscript

Accepted. Great suggestion. We have added a multiplication dot between L and b throughout for clarity.

R3-C2: 2) Equation 2 can only be possible if Q_p equals Q_s . This is already stated as (“assumed equal over the annual time period”), but when I read that the first time, I read it as no variation in time within a year which is confusing. Maybe change the sentence to: “assumed equal over the annual time period, so that $Q_p=Q_s$ ”.

Accepted. This was revised following your suggestion (Line 143).